# TAMING DATA AND TRANSFORMERS FOR AUDIO GENERATION

## ABSTRACT

Generating ambient sounds is a challenging task due to data scarcity and often insufficient caption quality, making it difficult to employ large-scale generative models for the task. In this work, we tackle this problem by introducing two new models. First, we propose AutoCap, a *high-quality* and *efficient* automatic audio captioning model. By using a compact audio representation and leveraging audio metadata, AutoCap substantially enhances caption quality, reaching a CIDEr score of 83.2, marking a 3.2% improvement from the best available captioning model at *four times* faster inference speed. Second, we propose GenAu, a scalable transformer-based audio generation architecture that we scale up to 1.25B parameters. Using AutoCap to generate caption clips from existing audio datasets, we demonstrate the benefits of data scaling with synthetic captions as well as model size scaling. When compared to state-of-the-art audio generators trained at similar size and data scale, GenAu obtains significant improvements of 4.7% in FAD score, 22.7% in IS, and 13.5% in CLAP score, indicating significantly improved quality of generated audio compared to previous works. Moreover, we propose an efficient and scalable pipeline for collecting audio datasets, enabling us to compile 57M ambient audio clips, forming AutoReCap-XL, the *largest* available audio-text dataset, at 90 times the scale of existing ones. Our code, model checkpoints, and dataset will be made publicly available upon acceptance.

## 1 INTRODUCTION

Text-conditioned generative models have revolutionized the field of content creation, enabling the generation of high-quality natural images (Ramesh et al., 2022; Rombach et al., 2022; Podell et al., 2023; Haji-Ali et al., 2024), vivid videos Ho et al. (2022); Villegas et al. (2022); Wang et al. (2023b); Qiu et al. (2023); Menapace et al. (2024), and intricate 3D shapes (Cheng et al., 2023). The domain of audio synthesis has undergone comparable advancement (Huang et al., 2023b;a; Liu et al., 2023b; Xue et al., 2024; Guan et al., 2024; Saito et al., 2024; Niu et al., 2024; Yang et al., 2023a; Evans et al., 2024b; Liu et al., 2024; Wang et al., 2024c; Guo et al., 2023), with three broad areas of study: speech, music and ambient sounds. The success in these domains rests on two key pillars: (i) the availability of high-quality large-scale datasets with text annotations, and (ii) the development of scalable generative models (Ho et al., 2020; Song et al., 2020).

In the field of audio synthesis, ambient audio generation emerges as a critical domain, which is the main focus of this work. Unlike speech and music, ambient sound generation suffers from a lack of extensive, well-annotated datasets (Kim et al., 2019; Drossos et al., 2020). Attempts to curate ambient audio from online videos predominantly failed, primarily due to the dominance of speech and music content in such videos. For instance, AudioSet (Gemmeke et al., 2017), the largest available audio dataset sourced from online videos, contains 99% speech or music clips. Previous efforts to filter ambient audio from similar datasets involved using expensive classifiers on the video or audio content, making it impractical to compile a large-scale dataset due to the high rejection rate. In this work, we propose a simple, yet scalable filtering approach that leverages existing automatic video transcription to identify segments with ambient sounds. This method is not only more efficient but also more feasible, as it eliminates the need to download the audio or video content. Additionally, by using time-aligned transcripts, we reduce the rejection rate to only 83%. Through this approach, we built AutoReCap-XL, a dataset containing 57 million ambient audio clips sourced from existing video datasets, representing a 90-fold increase over the size of previously available datasets.

Another challenge in compiling large-scale text-audio datasets is providing accurate textual descriptions. For visual modalities, such as images and videos (Xue et al., 2022; Miech et al., 2019), researchers often relied on the raw description and metadata to train strong visual-text models including reliable captioners (Chen et al., 2024b). Similarly, speech modality benefits from strong automatic transcription models to provide textual annotations. For ambient sounds, however, the task is substantially more challenging as accompanying raw text tends to describe visual information or convey feelings, rather than detailing the audio content. Moreover, human-captioned audio datasets are limited, containing fewer than $51k$ text-audio pairs in total. This significantly impacts the training of current captioning models, making them more susceptible to overfitting and reducing their ability to generalize effectively. In this work, we address this challenge by introducing AutoCap, an efficient and high-quality audio captioning model that leverages visual information to enhance captioning.

AutoCap refines the commonly used encoder-decoder design based on a pretrained BART (Lewis et al., 2020) model by learning an intermediate representation using a Q-Former (Li et al., 2023a) module. By learning an intermediate representation, AutoCap provides better alignment between the encoded audio and the original BART token representation due to the Q-Former additional capacity compared to simple projection layers used in previous work (Kim et al., 2024b). Second, we propose to use metadata and captions derived from video content to aid the captioning process and in this way, remedy the data scarcity problem. Critically, we augment the encoder inputs to assume both audio features and a set of descriptive textual metadata including audio title and a caption derived from the visual modality. This dual-input approach not only allows our model to achieve state-of-the-art performance on AudioCaps (Kim et al., 2019), marking a 3.2% improvement in CIDEr score, but it also helps reduce the domain gap with in-the-wild audios.

Moreover, to adapt audio generative models for larger scale training, we introduce GenAu, a scalable transformer-based architecture that achieves *significant* improvements over state-of-the-art audio generation models. Our approach introduces key architectural modifications over existing audio latent diffusion models (Liu et al., 2023b; Huang et al., 2023b; Ghosal et al., 2023; Huang et al., 2023a). First, we train an efficient 1D-VAE (Huang et al., 2023a) to transform a Mel-Spectrogram representation to a sequence of tokens and search for the optimal latent space for audio generation. Second, we recognize that audio grows fast temporally, yet contains many silent and redundant segments. Therefore, an efficient architecture that can handle such properties is needed. In particular, we employ a transformer architecture in the denoising backbone where we modify the FIT transformer (Chen & Li, 2023) to generate audio in the latent space. Lastly, we extend the proposed FIT architecture to incorporate text conditioning through a dual encoder strategy. This involves an instruction-finetuned language model, FLAN-T5 (Chung et al., 2022), and an audio-centric CLAP encoding (Wu et al., 2023a). This adaptations significantly improves the model's performance over exiting methods, achieving 22.7% higher Inception Score, 4.7% better FAD, and 13.5% improvement in CLAP score, demonstrating superior audio-text alignment and audio generation quality.

Finally, we explore the scaling behavior of text-to-audio diffusion models in relation to model size and data size. While text-to-image studies have shown performance improvements with increased data and model size (Peebles & Xie, 2023b; Li et al., 2024), similar exploration for audio remains limited. Initially, we analyze the impact of augmenting the dataset with synthetic captions on model performance. Our findings reveal a clear trend of improvement in FD and IS as we increase the amount of training data. Furthermore, we observe a consistent trend of enhanced performance across all metrics when scaling up the model size, concluding that the audio modality also benefits significantly from increases in both model size and data scale.

In summary, this work introduces: (i) AutoCap, a state-of-the-art audio captioner tailored towards the annotation of data at a large scale, which leverages audio metadata to improve accuracy and robustness, and a Q-Former to improve inference time and reduce overfitting; (ii) GenAu, a novel audio generator based on a scalable transformer architecture specifically adapted to the audio domain. Our model achieves significantly improved quality when compared to the previous state-of-the-art. (iii) AutoReCap-XL, the largest available audio dataset, comprising 57M audio clips paired with synthetic captions derived from the proposed audio captioner.

## 2 RELATED WORK

**Automatic Audio Captioning (AAC).** The goal of AAC is to produce natural language descriptions for given audio content. Most recent AAC methods (Deshmukh et al., 2023a; Wu et al.,

2024) (Salewski et al., 2023; Sridhar et al., 2023; Kadlčík et al., 2023; Cousin et al., 2023; Labbé et al., 2023; Xu et al., 2023; Zhang et al., 2024; Ghosh et al., 2024a; Deshmukh et al., 2023b) employ encoder-decoder transformer architectures, where an encoder receiving the audio signal produces a representation that is used by the decoder to produce the output caption. WavCaps (Mei et al., 2023a) employs the CNN14 (Kong et al., 2019) and HTSAT (Chen et al., 2022) audio encoders and uses a pretrained BART (Lewis et al., 2020) language decoder. CoNeTTE (Étienne Labbé et al., 2023) proposes an audio encoder based on the ConvNeXt architecture and uses a vanilla transformer decoder (Vaswani et al., 2017) trained from scratch. Recently, EnCLAP (Kim et al., 2024b) proposes the joint use of two audio representations in the form of CLAP (Wu et al., 2023a) sequence embeddings and a discrete EnCodec (Défossez et al., 2022) audio representation and uses a pretrained BART model as the language backbone. Other work explores augmentation strategies to counter data scarcity (Kim et al., 2022; Étienne Labbé et al., 2023; Ye et al., 2022). More recent work (Liu et al., 2023d; Sun et al., 2024; Yuan et al., 2024) proposed to leverage the visual information using to address sound ambiguities, reporting improvements. BART-Tags (Gontier et al., 2021) generates captions conditioned on a sequence of predicted AudioSet tags. Our method uses audio metadata and visual information as additional conditioning signals and leverages a lightweight Q-Former (Li et al., 2023a) model that summarizes the audio feature to improve captioning speed and reduce overfitting.

**Text-conditioned audio generation.** The current state-of-the-art text-to-audio generation methods widely adopt diffusion models (Yang et al., 2023b; Kreuk et al., 2023; Liu et al., 2023b;c; Huang et al., 2023a; Ghosal et al., 2023; Evans et al., 2024a; Vyas et al., 2023; Kreuk et al., 2023). AudioLDM (Liu et al., 2023b) makes use of a latent diffusion model conditioned on CLAP embeddings, reducing the need for the textual modality at training time. AudioLDM 2 (Liu et al., 2023c) introduces a general representation of audio unifying the tasks of music, speech, and sound effects generation. Similarly, Audiobox (Vyas et al., 2023) generates audio across different modalities such as speech and sound effects. Recently, StableAudio Open (Evans et al., 2024c) introduced a 1.32B model that uses a DiT (Peebles & Xie, 2023a) to generate variable-length audio clips at 48kHz. Recent work also explored controllable audio generation (Shi et al., 2023; Xu et al., 2024; Melechovsky et al., 2024; Paissan et al., 2024; Zhang et al., 2023b; Liang et al., 2024; Liu et al., 2023a), visual-conditioned audio generation (Wang et al., 2024d; Mei et al., 2023b; Wang et al., 2023a), and more recently joint audio-video generation (Tang et al., 2023a;b; Xing et al., 2024; Hayakawa et al., 2024; Tian et al., 2024; Vahdati et al., 2024; Chen et al., 2024a; Kim et al., 2024a; Wang et al., 2024a; Mao et al., 2024; Chen et al., 2024c). In this work, we show that improvements to data captioning quality and size, and the adoption of scalable architecture designs lead to state-of-the-art generation performance.

**Text-Audio Datasets.** The performance of text-audio models (Zhu et al., 2024; Li et al., 2023b; Deshmukh et al., 2024a; Mahfuz et al., 2023; Deshmukh et al., 2024c; Shu et al., 2023; Elizalde et al., 2024; Liu et al., 2023f; Tang et al., 2024; Gong et al., 2024a; Cheng et al., 2024; Zhang et al., 2023a), including AAC, is currently hindered by the lack of high-quality large-scale paired audio text data of ambient sounds. The two main existing datasets are AudioCaps (Kim et al., 2019) and Clotho (Drossos et al., 2020), comprising only $46k$ and $6k$ respectively of human-captioned audio clips. Another challenge is the limited availability of audio clips from sound-only platforms. LAION-Audio (Wu et al., 2023a) relied on numerous sources of audio platforms such as BBC Sound Effects (BBC Sound Effects, 2024), (Font et al., 2013) FreeSounds, and SoundBible (SoundBible, 2024) to form a dataset consisting of 630k audio samples with highly noisy raw descriptions. WavCaps (Mei et al., 2023a) proposes a filtering procedure based on ChatGPT (Achiam et al., 2023) to collect $400k$ audio clips and weakly caption them based on the noisy descriptions alone. Several subsequent work (Majumder et al., 2024; Sun et al., 2024) adopted similar strategies of using large language models to augment captions. While weak-captioning does improve downstream metrics, it is suboptimal because it fails to incorporate the audio signal itself. A recent work (Huang et al., 2023b) explored a knowledge distillation approach that leverages data labels and a pre-trained audio captioner and retriever to improve caption quality. Chen et al. (2020) attempted to extract audio clips from videos by employing classifiers to detect ambient audio, speech, and music. In this work, we introduce an efficient dataset collection pipeline that relies on video datasets to extract ambient audio clips. We use this approach to collect $57M$ audio clips and use our state-of-the-art captioning method to add audio-aligned text descriptions, compromising the largest available text-audio-video dataset.

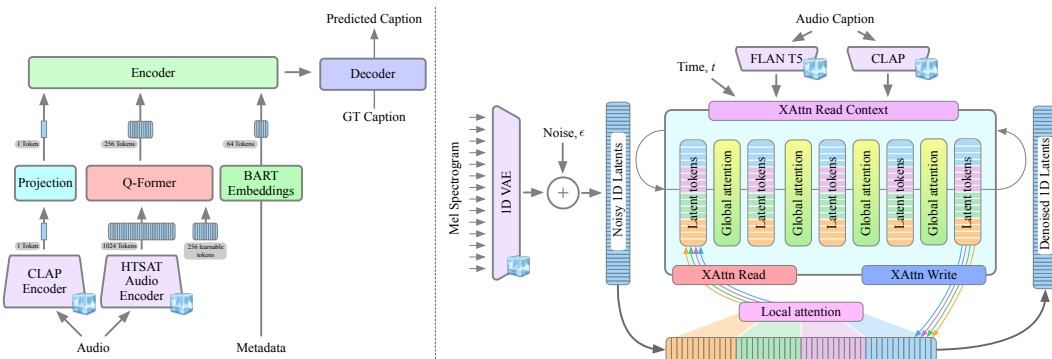

Figure 1: **(Left) Overview of AutoCap.** We employ a frozen HTSAT (Chen et al., 2022) encoder to produce an audio representation of 1024 tokens. We then employ a Q-Former (Li et al., 2023a) module to produce a 256 tokens. This representation, along with projected audio embeddings derived from a frozen CLAP audio encoder (Wu et al., 2023a) and 64 tokens derived from pertinent metadata, is processed by a pretrained BART encoder-decoder model to generate the final caption. **(Right) Overview of GenAu.** Following latent diffusion models, we use a frozen 1D-VAE to convert a Mel-Spectrogram into latent sequences, which are then divided into groups and processed using 'local' attention layers based on the FIT architecture (Chen & Li, 2023). 'Read' and 'write' layers, implemented as cross-attention, facilitate information transfer between input latents and *learnable* latent tokens. Finally, 'global' attention layers on *latent tokens* allow for global communication across all groups.

## 3 METHOD

In this section, we describe our approach to high-quality text-to-audio generation, starting with audio captioning using AutoCap in section 3.1, data collection and processing in section 3.2, and ambient audio generation with GenAu in section 3.3

### 3.1 AUTOMATIC AUDIO CAPTIONING

Audio is an inherently ambiguous modality, as many events can produce similar sound effects—a phenomenon often leveraged in animation, where soundscapes are artificially constructed. AAC attempts to generate textual descriptions for audio clips. Previous AAC methods have generally adopted an encoder-decoder transformer design, where an audio encoder is responsible for producing a representation that is processed by the decoder to produce a caption. Recent state-of-the-art methods (Étienne Labbé et al., 2023; Kim et al., 2024b) employ a pretrained audio encoder and finetune a pre-trained language model as the decoder, relying solely on the audio content for captioning. We believe that this approach is suboptimal. By directly finetuning the pre-trained language model on the limited available dataset, these methods often suffer from overfitting and limited expressiveness and accuracy. Audio files from many sources, however, are still commonly associated with metadata that might be relevant for captioning such as raw user descriptions, or a related modality (*i.e.* accompanied visual information). Motivated by this observation, we propose AutoCap, an audio captioning model that employs an intermediate audio representation to connect the pretrained encoder and decoder, and uses metadata to aid with the audio captioning. Figure 1 (left) presents an overview of AutoCap.

We consider a dataset of audio signals paired with a corresponding caption $\langle \mathbf{a}, \mathbf{y} \rangle$ and metadata represented as a set of token sequences $\{\mathbf{m}_j\}_{j=1}^{j=M}$. Inspired by state-of-the-art AAC methods (Mei et al., 2023a; Étienne Labbé et al., 2023; Kim et al., 2024b), we employ an encoder-decoder architecture. We start by computing a global feature representation of the audio:

$$\mathbf{x}_{\text{clap}} = \mathcal{P}_{\text{clap}}(\mathcal{E}_{\text{clap}}(\mathbf{a})), \tag{1}$$

where $\mathcal{P}_{\text{clap}}$ is a learnable projection layer and $\mathcal{E}_{\text{clap}}$ is the audio encoder of a pretrained CLAP model (Wu et al., 2023a). Then we compute a local feature representation of the input audio:

$$\mathbf{x}_{\text{audio}} = \mathcal{Q}(\mathcal{E}_{\text{a}}(\mathbf{a})), \tag{2}$$

Figure 2: **Audio data collection pipeline.** We employ online video transcripts to identify audio segments without speech or music. These are processed by AutoCap to generate captions. We retain only ambient clips with captions lacking music and speech keywords.

where $\mathcal{Q}$ is a Q-Former (Li et al., 2023a) that outputs a compact sequence audio representation and $\mathcal{E}_a$ is a pretrained HTSAT (Chen et al., 2022) audio encoder that produces a time-aligned representation. The Q-Former efficiently learns 256 latent tokens, which serve as keys in cross-attention layers with the input features, thereby condensing the audio input features into 256 tokens. Metadata sequences $\mathbf{m}_i$ are then embedded using the embedding layer of the pretrained decoder model to obtain corresponding embedding sequences $\mathbf{x}_{\text{meta}_i}$. For our experiments, we use video titles and captions as the metadata. We represent the input audio and metadata as the following input sequence:

$$\mathbf{x} = \mathbf{x}_{\text{clap}} \; [\texttt{boa}] \; \mathbf{x}_{\text{audio}} \; [\texttt{eoa}] \; [\texttt{bom}]_1 \; \mathbf{x}_{\text{meta}_1} \; [\texttt{bom}]_1 \; ... \; [\texttt{bom}]_M \; \mathbf{x}_{\text{meta}_M} \; [\texttt{bom}]_M, \quad (3)$$

where $[\texttt{boa}] [\texttt{eoa}]$ represent beginning and end of audio sequence embeddings $\mathbf{x}_{\text{audio}}$, and $[\texttt{bom}]_i, [\texttt{bom}]_i$ represent beginning and end of metadata embeddings $\mathbf{x}_{\text{meta}_i}$. This input sequence is then used to obtain an output predicted caption $\hat{\mathbf{y}}$ as $\hat{\mathbf{y}} = \mathcal{D}_t(\mathbf{x})$ where $\mathcal{D}_t$ is a pretrained BART transformer model (Lewis et al., 2020) serving as the decoder. Finally, we train our model using a standard cross-entropy loss over next token predictions:

$$\mathcal{L}_{\text{CE}} = -\frac{1}{T} \sum_{t=1}^{t=T} \log p(\mathbf{y}_t | \mathbf{y}_{1:t-1}, \mathbf{x}). \quad (4)$$

To avoid degrading the quality of the pretrained BART and audio encoder models, we adopt a two-stage training procedure. In Stage 1, both the audio encoders and BART model are kept frozen, thus allowing the Q-Former, projection layers, and newly introduced delimiter tokens to align to the existing BART input representation. In this stage, we pretrain the model using a larger dataset of weakly-labeled audio clips. In Stage 2, we unfreeze all BART model parameters apart from the embedding layer and finetune the model on the Audiocaps dataset at a lower learning rate to make the captioning style align more closely to the target dataset. This training strategy effectively leverages the larger, weakly-labeled dataset while minimizing the knowledge drift in the pretrained BART. The use of Q-Former to learn an intermediate representation is pivotal for such training strategy. Furthermore, compared with baseline HTSAT-BART (Mei et al., 2023a), the Q-Former summarizes the audio representation into four times fewer tokens, significantly reducing the inference time.

## 3.2 DATA COLLECTION AND RE-CAPTIONING PIPELINE

Generative models in the image and video domains have shown benefits from increased quantities of data and improved quality of captions. In the audio domain, however, the major human-annotated audio-text datasets, namely AudioCaps (Kim et al., 2019) and Clotho (Drossos et al., 2020), provide only 51k audio clips combined. Previous methods attempted to extract additional ambient audio clips from existing video datasets using pretrained audio classifiers, but a high rejection rate of 99% marked this method impractical. Instead, we found that automatic transcripts offer reliable information about the segments containing ambient sounds. In particular, we propose to select the parts of the videos that contain no automatic transcription, suggesting the absence of speech and music. Such an approach offers specific advantages over using pretrained classifiers. Automatic

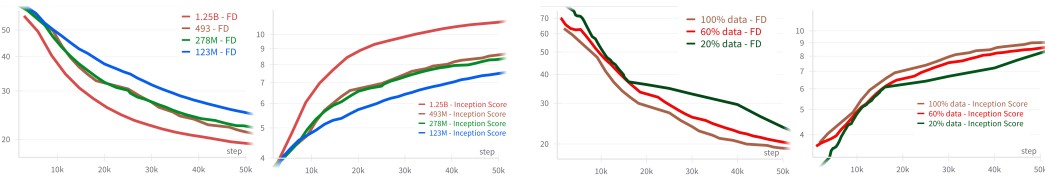

Figure 3: **Scaling analysis** of model size (left) and data volume with synthetic captions (right) reveal consistent improvements in FD and IS.

transcripts, readily available for most online videos, eliminate the need to download and process video and audio data before filtering. Additionally, as these transcripts provide precise time-aligned information, they facilitate the extraction of more segments, effectively reducing the rejection rate to 83%. Subsequently, we leverage our AutoCap model to provide textual descriptions of the extracted audio clips. Despite the effectiveness of this method in collecting ambient sounds, some clips still inadvertently contain music or speech due to transcription errors, particularly with speech in less common languages. We address this by analyzing captions and filtering out clips with keywords related to speech or music. Figure 2 summarizes our data collection pipeline.

We follow this process to extract 466k audio-text pairs from Audioset (Gemmeke et al., 2017) and VGGSounds (Chen et al., 2020). Additionally, we recaption audio-only dataset such as Freesound, BBC Sound Effects, and SoundBible. To provide metadata, we employ the captioning model of Chen et al. (2024b) to extract a caption whenever a video content is available and pass an empty text otherwise. In total, we form AutoReCap, a large-scale dataset compromising of *761,113* audio-text pairs with precise captions. As an additional contribution, we introduce AutoReCap-XL, in which we scale our approach by analysing four additional large-scale video dataset (Lee et al., 2021; Xue et al., 2022; Zellers et al., 2022; Nagrani et al., 2022) with a total of *71M* videos and $715.4k$ hours. In total, we collect and re-caption *57M* ambient audio clips spanning $123.5k$ hours from *20.3M* different videos, forming by far the largest available dataset of audio with paired captions. More details about the dataset can be found in the *Appendix*.

## 3.3 SCALABLE TEXT-2-AUDIO GENERATION

We design our audio generation pipeline, GenAu, as a latent diffusion model. Figure 1 (right) shows an overview of our proposed model. In the following section, we describe in detail the structure of our latent variational autoencoder (VAE) and the latent diffusion model.

**Latent VAE.** Directly modeling waveform audio data is complex due to the high data dimensionality of audio signals. Instead, we replace the waveform with a Mel-spectrogram representation and use a VAE to further reduce its dimensionality, following prior work (Melechovsky et al., 2024; Huang et al., 2023b). Once generated, Mel-spectrograms can be decoded back to a waveform through the use of an audio vocoder (Kong et al., 2020). However, commonly-used 2D autoencoder designs (Liu et al., 2023b;c; Melechovsky et al., 2024), are not well suited to the Mel-spectrogram representation, as the separation between the Mel channels is non-linear, which is not well suited for 2D convolutions. We instead opt for a 1D-VAE design based on 1D convolutions similar to Huang et al. (2023a). We train our VAE using a combination of reconstruction, adversarial, and KL regularization losses following Esser et al. (2021).

**Latent diffusion model.** Following the latent diffusion paradigm, we generate audio by training a diffusion model in the latent space of the 1D-VAE. Transformer-based diffusion models currently attain state-of-the-art performance in audio generation (Huang et al., 2023a). To improve model scalability, we propose to use an efficient transformer architecture due to its success in handling long-range interactions as in video generation (Chen & Li, 2023; Menapace et al., 2024). In particular, we adopt the FIT architecture of Menapace et al. (2024) which was originally proposed to work in the *pixel space* and revise it for the *latent space* of the audio modality.

Given a 1D input $\mathbf{x}$, we first apply a projection operation to produce a sequence of input patch tokens. We then apply a sequence of FIT blocks to the input patches where each block divides patch tokens into contiguous groups of a predefined size. A set of *local* self-attention layers are then applied separately to each group to avoid the quadratic computational complexity of attention computation. Differently from the video domain (Menapace et al., 2024) where the high input dimensionality

Table 1: AutoCap results on AudioCaps test split for various models. AS: AudioSet, AC: AudioCaps, WC: WavCaps, CL: Clotho, MA: Multi-Annotator Captioned Soundscapes.

| Model | Pretraining Data | BLEU1 | BLEU4 | ROUGE$_L$ | METEOR | CIDEr | SPICE | SPIDEr |
|---|---|---|---|---|---|---|---|---|
| ACT | AS | 64.7 | 25.2 | 46.8 | 22.2 | 67.9 | 16.0 | 42.0 |
| V-ACT | - | 69.8 | 28.1 | 49.4 | 23.7 | 71.1 | 17.2 | 44.2 |
| BART-tags | AS | 69.9 | 26.6 | 49.3 | 24.1 | 75.3 | 17.6 | 46.5 |
| AL-MixGEN | - | 70.0 | 28.9 | 50.2 | 24.2 | 76.9 | 18.1 | 47.5 |
| ENCLAP-Large | - | - | - | - | 25.5 | 80.2 | 18.8 | 49.5 |
| HTSAT-BART | - | 67.5 | 27.2 | 48.3 | 23.7 | 72.1 | 16.9 | 44.5 |
| HTSAT-BART | AC+CL+WC | 70.7 | 28.3 | 50.7 | 25.0 | 78.7 | 18.2 | 48.5 |
| CNext-trans | - | - | - | - | - | - | - | 46.6 |
| CNext-trans | AC+CL+MA+WC | - | - | - | 25.2 | 80.6 | 18.4 | 49.5 |
| AutoCap (audio) | AC | 70.0 | 28.0 | 51.7 | 24.6 | 77.3 | 18.2 | 47.8 |
| AutoCap (audio+text) | AC | 72.1 | 28.6 | 51.5 | **25.6** | 80.0 | 18.8 | 49.4 |
| AutoCap (audio) | AC+CL+WC | **73.1** | 28.1 | **52.0** | **25.6** | 80.4 | **19.0** | 49.7 |
| AutoCap (audio+text) | AC+CL+WC | 72.3 | **29.7** | 51.8 | 25.3 | **83.2** | 18.2 | **50.7** |

makes the *local* layers excessively expensive, we found them to be beneficial for audio generation. To further reduce the amount of computation while maintaining long-range interaction, each block considers a small set of latent tokens. First, a *read* operation implemented as a cross-attention layer transfers information from the patches to the latent tokens. Later, a series of *global* self-attention operations are applied to the latent tokens, allowing information-sharing between different groups. Finally, a *write* operation implemented as a cross-attention layer transfers information from the latent tokens back to the patches. Due to the reduced number of latent tokens when performing the global self-attention, computational requirements of the model are reduced with respect to a vanilla transformer design (Vaswani et al., 2017). Such a design is also particularly suited for the audio modality, which contains mostly silent or redundant parts. Unlike DiT and UNet-based methods (Ronneberger et al., 2015; Peebles & Xie, 2023b) which allocate the computation resources uniformly across input tokens, the FiT architecture selectively focuses on the more informative parts.

To condition the generation on an input prompt, we use a pretrained FLAN-T5 model (Chung et al., 2022) and a CLAP (Wu et al., 2023a) text encoder to produce the their respective embeddings $e_{\text{FLAN}}$ and $e_{\text{CLAP}}$ following prior work Liu et al. (2023c), which we concatenate with the diffusion timestep $t$ to form the input conditioning signal $c$. We insert an additional cross-attention operation inside each FIT block immediately before the 'read' operation that makes latent tokens attend to the conditioning. Moreover, we use conditioning on dataset ID to adapt the generation style to different datasets.

We follow a linear noise scheduler and train the model using the epsilon prediction objective:

$$\mathcal{L} = \mathbb{E}_{t,\mathbf{x},\boldsymbol{\epsilon}} \left\| \mathcal{G}(\mathbf{x}_t, c) - \boldsymbol{\epsilon} \right\|_2^2, \tag{5}$$

where $\mathcal{G}$ is the FIT generator backbone, $\mathbf{x}_t$ is the input with applied noise at diffusion timestep $t$, and $\boldsymbol{\epsilon}$ is noise sampled in $N(0, 1)$ with the same shape as the input.

## 4 EXPERIMENTS

We structure the experiments section as follows: section 4.1 evaluates AutoCap by quantitatively comparing it to previous work, section 4.2 demonstrates the capabilities of GenAu quantitatively. For both, we discuss training details, baselines, metrics, results, and ablations.

### 4.1 AUTOMATIC AUDIO CAPTIONING

**Training dataset and details.** We train AutoCap in two stages. During stage 1, we pretrain on a large weakly labeled dataset of 634,208 audio clips, constructed from AudioSet, Freesound, BBC Sound Effects, SoundBible, AudioCaps, and Clotho. We use ground truth captions from AudioCaps and Clotho dataset, WavCaps captions for Freesound, SoundBible, and BBC Sound Effects, and

Table 2: AutoCap ablation study on AudioCaps.

| Model | METEOR ↑ | CIDEr ↑ | SPICE ↑ | SPIDEr ↑ |
|---|---|---|---|---|
| Ours | **25.3** | **83.2** | 18.2 | **50.7** |
| - w/o CLAP | **25.3** | 80.7 | **18.4** | 49.6 |
| - w/o Stage 2 | 24.2 | 75.6 | 17.3 | 46.5 |
| - w/o Stage 1 | 22.6 | 59.6 | 15.4 | 37.5 |
| - Unfreeze Word Embed | 22.5 | 82.6 | 18.1 | 50.4 |

Table 3: GenAu ablation study on out-of-distribution dataset.

| Model | IS | FD | CLAP$_{MS}$ |
|---|---|---|---|
| GenAU-L | **18.98** | **20.81** | **0.38** |
| GenAU-L (AC) | 12.14 | 25.82 | 0.30 |
| GenAU-S | 15.76 | 21.29 | 0.36 |
| GenAU-S w/o Recap. | 11.83 | 25.34 | 0.29 |

handcrafted captions through a template leveraging the ground truth class labels for AudioSet. As metadata, we use the title provided with each clip, and pre-compute video captions using a pretrained Panda70M model (Chen et al., 2024b) when the video modality is available or pass an empty string otherwise. We pretrain the model for 20 epochs with a learning rate of 1e-4, while keeping the audio encoder and pretrained BART frozen. In Stage 2, we fine-tune the model for 20 epochs on AudioCaps using a learning rate of 1e-5. We use 10-second clips at 32KHz for all experiments.

**Baselines.** We compare with ACT (Mei et al., 2021), V-ACT (Liu et al., 2023e), BART-tags (Gontier et al., 2021), AL-MixGEN (Kim et al., 2022), ENCLAP (Kim et al., 2024b), HTSAT-BART (Xu et al., 2023) and CNext-trans (Étienne Labbé et al., 2023). Among these baselines, ENCLAP and CNext-trans achieve the best performance. ENCLAP benefits from a stronger audio encoder and the use of a CLAP representation for additional guidance. CNext-trans trains a lightweight transformer instead of fine-tuning a pretrained language model to reduce overfitting.

**Metrics and evaluation.** We report results using the the established BLEU1 (Papineni et al., 2002), BLEU2 (Papineni et al., 2002), ROUGE (Lin, 2004), Meteor (Lavie & Agarwal, 2007), CIDEr (Vedantam et al., 2015), and SPIDEr (Liu et al., 2017) metrics. We evaluate our method on the AudioCaps test split using the last checkpoint of our trained model. We used only 876 clips for evaluation as some videos were deleted since the original data release. We follow the same evaluation pipeline as baselines and include their reported results. Results that were not provided in these publications are excluded from our analysis.

**Results.** In Tab. 1 we report the quantitative comparison. Our method outperforms previous methods on all metrics, achieving notable improvements in the CIDEr and BLUE1 scores, with values of 83.2 and 73.1, respectively. We found that incorporating metadata significantly enhances the CIDEr scores but slightly reduces the SPICE scores. This trade-off likely results from the enhanced descriptive detail brought by the metadata, which while enriching the content, introduces noise that may compromise the model's semantic precision. In addition, AudioCaps is labeled based on audio information alone. Thus, the evaluation penalizes the description of information that can not be deduced with certainty from the audio modality only, such as the specific type of object producing a rustling sound. Compared to ENCLAP-Large (Kim et al., 2024b), and CNext-trans (Étienne Labbé et al., 2023), we find the captions produced by our method to be more descriptive and precise with a better temporal understanding. ENCLAP-Large often misses important details and exhibits lower temporal accuracy. CNext-trans, while accurate, often produces short captions that lack details. We include qualitative comparisons in the project *Website*. Moreover, AutoCap is *four times* faster than ENCALP, producing a caption for a 10-second clip in *0.28* seconds, compared to ENCALP which takes *1.12* seconds. Furthermore, we observe consistent improvements when pretraining on a large scale of weakly-labeled data during the first stage, validating the effectiveness of our training strategy in benefiting from a larger, weakly-labeled dataset.

**Ablations.** In Tab. 2, we ablate model design choices. We observe the use of the CLAP embedding to bring a 2.5 points increase in the CIDEr score. We also validate that when not performing Stage 2 training, which involves finetuning of the BART (Lewis et al., 2020) model, performance degrades on all metrics, a finding we attribute to the necessity of adapting BART's decoder to the sentence structure typical of AudioCaps. A more severe degradation in performance is observed if Stage 1 is not performed, with the misaligned representation between the encoder and the decoder causing catastrophic forgetting in the language model. Finally, if BART word embeddings are finetuned in Stage 2 instead of being kept frozen, we observe a slight performance degradation.

## 4.2 TEXT-2-AUDIO GENERATION

**Training dataset and details.** We train on similar data settings to baselines. We use our best-performing captioning model to re-caption the WavCaps dataset. In addition, we obtain 339,387

Table 4: GenAu results on AudioCaps test split.

| Model | Prams | # Samples | FD ↓ | IS ↑ | FAD ↓ | CLAP$_{LAION}$ ↑ | CLAP$_{MS}$ ↑ |
|---|---|---|---|---|---|---|---|
| GroundTruth | - | - | - | - | - | 0.251 | 0.671 |
| AudioLDM-L | 739M | 634k | 37.89 | 7.14 | 5.86 | - | 0.429 |
| AudioLDM 2-L | 712M | 760k | 32.50 | 8.54 | 5.11 | 0.212 | 0.621 |
| TANGO | 866M | 45k | 26.13 | 8.23 | 1.87 | 0.185 | 0.597 |
| TANGO 2 | 866M | 60k | 19.77 | 8.45 | 2.74 | 0.264 | 0.590 |
| Make-An-Audio | 453M | 1M | 27.93 | 7.44 | 2.59 | 0.207 | 0.621 |
| Make-An-Audio 2 | 937M | 1M | **15.34** | 9.58 | 1.27 | 0.251 | 0.645 |
| Stable Audio Open | 1.32B | 486K | 21.23 | 10.48 | 2.32 | 0.246 | 0.584 |
| GenAu w/ U-Net | 462M | 811K | 25.57 | 9.54 | 1.98 | - | - |
| GenAu-Large | 1.25B | 811K | 16.51 | **11.75** | **1.21** | **0.285** | **0.668** |

videos from AudioSet and 126,905 videos from VGGSounds, totaling 761,113 clips. For those obtained from sound-only platforms, we input an empty string as the video caption. For full details of the data sources of our training dataset, please refer to the *Appendix*. We additionally use Clotho and AudioCaps training datasets with their ground truth caption. To stay consistent with baselines, we train at 16kHz resolution. We use a patch size of 1 and a group size of 32. We use LAMB optimizer (You et al., 2020) with a LR of 5e-3. We train for 220k steps and choose the checkpoint with the highest IS, at steps 210k and 207k for the large and small models. We also disable EMA as found it to make the metrics unstable.

**Baselines.** We compare with TANGO 1 & 2, (Ghosal et al., 2023), AudioLDM 1 & 2 (Liu et al., 2023b;c), and Make-An-Audio 1 & 2 (Huang et al., 2023b;a). Both AudioLDM and Make-an-Audio train a UNet-based latent diffusion model (Rombach et al., 2022) on Mel-Spectrogram representation of the audio, by regarding the Mel-Spectrogram as a single channel image, and use a pretrained CLAP encoder to condition the generation on an input prompt. TANGO proposed to use FLAN-T5 (Chung et al., 2022) as the text encoder and reported significant improvements. AudioLDM-2 and Make-an-Audio-2 proposed to use a dual encoder strategy of a T5 (Raffel et al., 2022) and CLAP encoder. AudioLDM-2 focused on extending the generation and conditioning to various domains. Specifically, they use the language of audio (LOA) to condition the generation on images, audio, or transcripts and train their model for music and speech generation. Make-an-Audio-2 proposes to use a 1D VAE representation and employ a feed-forward Transformer-based model to replace the UNet. Recently, Tango-2 proposed to use instruction fine-tuning on a synthetic dataset to enhance the temporal understanding. In our experiments, we focus on text-conditioned natural audio generation and generate 10s clips at a resolution of 16Khz.

Table 5: User study between various baselines. % of votes in favor of the baseline to the left.

| Model | Realism | Quality | Prompt Alignment | Overall Preference |
|---|---|---|---|---|
| GenAU-L vs GenAU-S | 61.20% | 58.00% | 61.20% | 60.40% |
| GenAU-L vs GenAU-L (AC) | 60.40% | 54.80% | 60.40% | 59.20% |
| GenAU-L vs MAD-2 | 64.00% | 62.40% | 68.40% | 66.40% |
| GenAU-S w/o Recap. vs MAD-2 | 64.40% | 64.00% | 63.20% | 64.80% |

**Metrics.** We compare the performance of our method with baselines using the standard Frechet Distance (FD), Inception score (IS), and CLAP score on the Audioset test dataset, containing 964 samples. There is little consistency between baselines when computing the metrics. Some prior work reported the Fréchet distance results using the VGGish network (Hershey et al., 2017), denoted as (FAD) (Kilgour et al., 2019), while other uses PANNs (Kong et al., 2019). Additionally, to compute the CLAP score, some prior work (Liu et al., 2023c) used CLAP from LAION, which we denote as CLAP$_{LAION}$ (Wu et al., 2023b), while others (Majumder et al., 2024; Huang et al., 2023b;a) used CLAP from Microsoft (Elizalde et al., 2023), which we denote as CLAP$_{MS}$. Furthermore, some prior (Liu et al., 2023b;c) used CLAP re-ranking with 3 samples for computing the metrics. Due

to such inconsistencies in evaluation pipelines and varying results for the same baselines reported in different studies, we recompute all metrics using the official checkpoints to ensure consistent comparisons. We follow the same evaluation protocols of AudioLDM (Liu et al., 2023b) without CLAP re-ranking and use the AudioLDM evaluation package to compute the metrics. Besides, we run our ablations on the Bigsoundbank split from WavText5k (Deshmukh et al., 2022), which serves as an out-of-distribution evaluation for our models. This is to prevent biasing the evaluation based on the training data. Finally, to further validate our results we run a user preference study. Details about the user study can be found in the *Appendix*.

**Results.** In Tab. 4, we report evaluation results. Our method achieves superior performance compared to the state-of-the-art methods in terms of IS, FAD, $CLAP_{MS}$ and $CLAP_{LAION}$ scores, marking an improvement of $22.7\%$, $4.7\%$, $3.6\%$, and $13.5\%$, respectively. This shows that GenAu can produce high audio quality and achieve better semantic alignment with the conditioning text.

**Data scaling.** We consider two key aspects: data quality and quantity. First, in Tab. 5 ($2^{nd}$ fow), we show that GenAu-L trained with AutoReCap is generally favoured over training only with AudioCaps (AC). This is confirmed in Tab. 3 ($1^{st}$ vs $2^{nd}$ row), where increasing the dataset size significantly boosts the results across all metrics, improving IS by $56.3\%$. Additionally, we show ($3^{rd}$ vs $4^{th}$ row) that using AutoCap to recaption the dataset significantly enhances the results over all metrics, confirming the importance of data quality. Interestingly, expanding the data size at a lower caption quality does not yield similar gains even at a bigger model ($2^{nd}$ vs $4^{th}$ row), aligning with results reported by Liu et al. (2023c). This highlights that data quality brought by AutoCap is as crucial as the data quantity. Lastly, we examine the effect of scaling the data with synthetic captions. For this, we train for 50k steps by fixing AC and Clotho in the training data and varying the amount of synthetic data. As reported in Fig. 3 (right), scaling data with synthetic caption has a clear improvement over both IS and FD, with the model trained on the whole AutoReCap achieving the best results.

**Model size scaling.** In Tab. 5, we report ($1^{st}$ row) that GenAu-L (1.25B params) is constantly favoured over GenAu-S (493M params). This is further confirmed by our automatic evaluation in Tab. 3 ($1^{st}$ vs $3^{rd}$ row), where the larger model shows significant improvements across all metrics. The scaling trend is also evident in Fig. 3, which demonstrates a clear correlation between model size and performance in terms of both IS and FD scores.

**Model architecture ablation.** Until recently, A UNet (Ronneberger et al., 2015) has been the most popular choice for the diffusion backbone. Yet, as reported in Tab. 4, replacing the FiT backbone with a UNet drastically reduces performance across all metrics. This supports baseline findings where UNet-based methods lag behind transformer-based approaches (Huang et al., 2023a). Another choice that has recently gained popularity is the DiT architecture (Peebles & Xie, 2023b). Make-an-Audio-2 (MAD-2) employs a DiT at a similar model size and data scale as GenAU-L. However, as we show in Tab. 5, our model is consistently preferred over MAD-2 ($3^{rd}$ row), even without dataset recaptioning ($4^{th}$ row) (*i.e.* at similar data settings). We infer that the FiT architect, with its read and write operations, allocates compute more efficiently to the key segments of the input, making it more suitable to ambient audio clips which often include silent or redundant parts.

## 5 CONCLUSION

We take a holistic approach to improve the quality of existing audio generators. Starting by addressing the scarcity of large-scale captioned audio datasets, we build a state-of-the-art audio captioning method, AutoCap, which leverages audio metadata to collect a dataset of 57M annotated audio clips. We then built a latent diffusion model based on a scalable transformer architecture which we trained on our re-captioned dataset to obtain GenAu, a state-of-the-art open-sources model for audio generation. Our approach not only improves ambient audio generation but also opens up possibilities for extending GenAu to other domains, such as speech and music generation. As an additional contribution, we built AutoReCap-XL, a text-audio-video ambient audio dataset with an unprecedented size of $57M$ pairs. AutoReCap-XL can potentially serve as a joint text-audio-video dataset and broadens novel applications such as text-to-audio-video joint generation.

**Limitations and future work.** AutoCap was fine-tuned on AudioCaps, featuring 4,892 unique words, which limits the diversity of our generated captions. Consequently, GenAu may face challenges in accurately generating audio for detailed prompts. While AutoReCap is extensive in size, it has only been validated for audio generation. We leave broader analysis on more tasks for future work.

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

# APPENDIX

## CONTENTS

## A   AUTORECAP-XL DETAILS

This section outlines the collection and filtering processes for AutoReCap-XL.

### A.1   STAGE 1: DATA SELECTION

We selected existing video datasets primarily from YouTube for the ease of accessing automatic transcriptions. Specifically, we chose 73 million videos from the datasets AudioSet (Gemmeke et al., 2017), VGGSound (Chen et al., 2020), ACAV100M (Lee et al., 2021), VideoCC (Nagrani et al., 2022), YTTEMP1B (Zellers et al., 2022), and HDVila-100M (Xue et al., 2022). We select these datasets for their likelihood of containing videos with strong audio-video correspondence.

### A.2   STAGE 2: SPEECH AND MUSIC FILTERING

We downloaded English transcripts from YouTube and used automatically generated ones for videos without existing transcripts. However, we discard videos without any transcripts. While some datasets provide only video segments with specific timestamps, we processed the full videos, totaling around 73 million videos. We accepted audio segments longer than one second that lacked any corresponding subtitles, indicating the absence of speech and music. After filtering, we isolated approximately 327.3 million segments from 55.1 million videos. Fig. 4 displays the distribution of the number of segments per video. We denote this dataset as AutoReCap-XL-Raw. Subsequently, we use AutoCap to caption the audio segments. Fig. 6 shows the distribution of caption lengths. Given that AutoCap was trained for 10-second audio, we limited segments to this duration. Additionally, we concatenate consecutive segments yielding identical captions to form longer audio clips. Fig. 8 illustrates the audio length distribution, and a word cloud of the captions is shown in Fig. 10. Despite filtering, the dataset was still dominated by captions related to speech and music. We attribute this to the limitations of YouTube's automatic transcription, particularly with certain types of music and less common languages.

### A.3   STAGE 3: POST-FILTERING OF SPEECH AND MUSIC.

To further refine the dataset from speech and music, We follow a simple filtering approach. Specifically, we employed a large language model (LLM) to generate keywords associated with speech

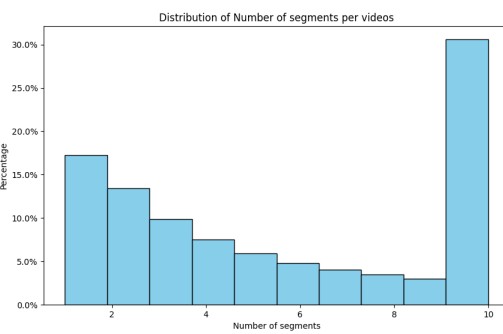

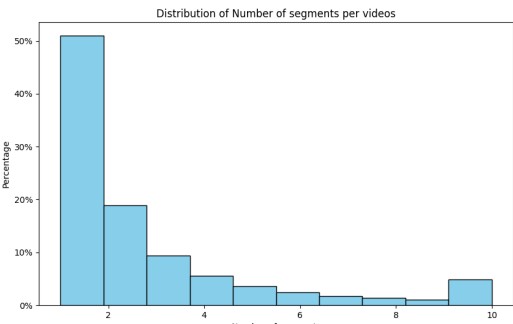

Figure 4: Distribution of number of segments per-video in AutoReCap-XL-Raw

Figure 5: Distribution of the number of segments per-video in AutoReCap-XL

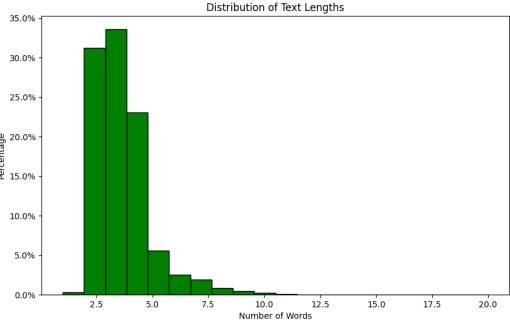

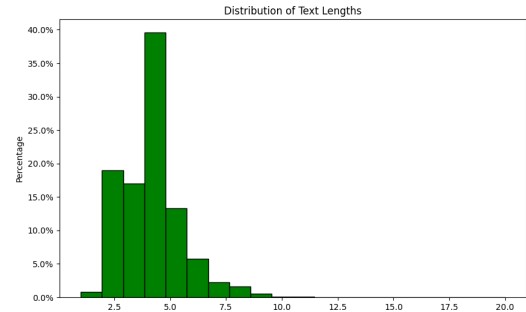

Figure 6: Distribution of caption length of AutoReCap-XL-Raw

Figure 7: Distribution of caption length of AutoReCap-XL

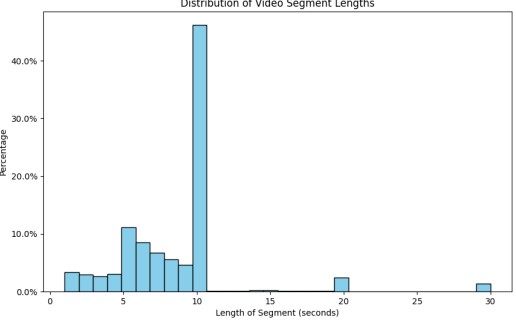

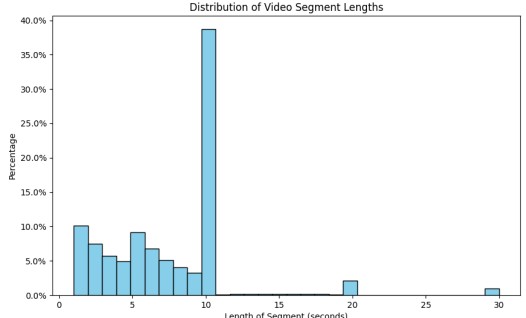

Figure 8: Distribution of audio duration of AutoReCap-XL-Raw

Figure 9: Distribution of audio duration of AutoReCap-XL

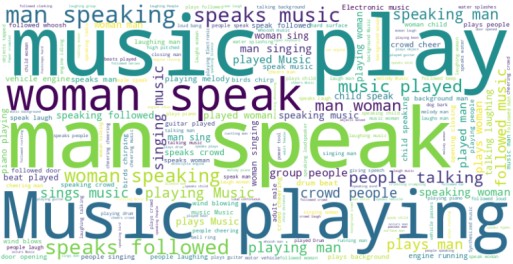

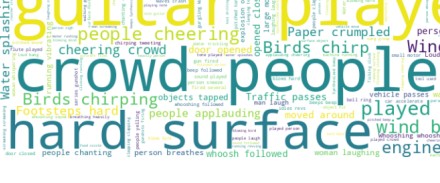

Figure 10: Word cloud of audio captions in AutoReCap-XL-Raw

Figure 11: Word cloud of audio captions in AutoReCap-XL

and music, such as "talking", "speaking", and "singing," and excluded all audio segments whose captions contained such keywords. This process yielded 57 million audio-text pairs from 20.3 million videos. Fig. 5 shows the number of segments per video, Fig. 7 shows the caption length distribution, Fig. 9 shows the audio length distribution, and Fig. 11 presents a word cloud of the final captions. We outline the data sources for constructing this dataset in Tab. 6. Our proposed dataset is not only 90 times larger than the previously largest available dataset, LAION-Audio-630KWu et al. (2023b) in terms of the number of audio clips, but also provides more accurate captions compared to existing datasets that rely on raw textual data. A comprehensive comparison with other datasets is detailed in Tab. 7

## B ARCHITECTURE DETAILS

### B.1 HTSAT EMBEDDINGS EXTRACTION

AutoCap uses HTSAT (Chen et al., 2022) embeddings to encode the input audio and follows the HTSAT-BART (Mei et al., 2023a) embedding extraction procedure, described in the following, to obtain "fine-grained" HTSAT embeddings. Given a 10-seconds single-channel input audio at 32Khz, HTSAT represents it as a mel-spectrogram using window size of 1024, 320 hop size, and 64 mel-bins, resulting in an input of shape ($T = 1024, F = 64$). The spectrogram is then encoded as latent tokens of shape ($\frac{T}{8P} = 32$, $\frac{F}{8P} = 2$, $8D = 768$) before the classification layer. HTSAT-BART Mei et al. (2023a), then averages over the frequency dimension to obtain a representation of shape ($\frac{T}{8P} = 32$, $1, 8D = 768$) and replicates the latent token by a token replication factor of $8P = 32$ to obtain a so-called "fine-grained" representation of shape $32 \times 32 \times 768$, which is flattened into a representation of shape $1024 \times 768$. We adopt this representation throughout our work, and Appx. G.3 provides additional evaluation results showing the performance benefits of the token replication operation.

## C LIMITATIONS

### C.1 AUTOCAP

Sounds emitted by various objects can often sound similar, such as a waterfall compared to heavy rain, or a can versus a motorcycle engine. In scenarios where metadata lacks detail, our audio captioning model may struggle to disambiguate these sounds accurately. The model also tends to falter in capturing the temporal relationships between sounds and differentiating foreground from background noises. Additionally, since it is fine-tuned on AudioCaps, which contains a limited vocabulary of 4,892 unique words (excluding common stop words), the model frequently produces repetitive words and captions.

### C.2 GENAU

Although our model is trained to generate natural sound effects, it underperforms in specialized areas like music generation or text-to-speech synthesis, where more targeted models are superior. Moreover, the limited vocabulary of the paired texts, even though extensive, hampers the model's ability to accurately generate audio for long and detailed prompts.

### C.3 AUTORECAP-XL

Our proposed dataset, AutoReCap-XL, is substantial in size but features a constrained vocabulary of only 4,461 unique words, excluding stop words, due to the vocabulary limitations of the AudioCaps-trained captioner. Furthermore, despite its potential as a significant contribution, this dataset has not yet been extensively analyzed for caption accuracy or performance in downstream tasks.

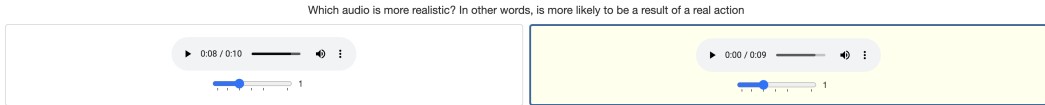

Figure 12: A screenshot of the user study interface.

## D EVALUATION DETAILS

### D.1 AUDIO CAPTIONING

While the established practice in the evaluation of audio captioning methods is to report the results on the test set using the checkpoint that performs best on the validation subset, prior work (Étienne Labbé et al., 2023; Kim et al., 2024b) reported high instability of the metrics on the validation subset and weak correlation between the validation and test performance, making the model's results vary significantly for different seeds. To alleviate this, ENCLAP (Kim et al., 2024b) selects around five best-performing validation checkpoints and reports their best results on the test set. CNext-trans (Étienne Labbé et al., 2023) uses the FENSE score to pick the best validation checkpoint. This method of choosing the best checkpoint may produce misleading results and potentially disadvantage baselines. Our model, thanks to the two-stage training paradigm, significantly reduces this instability and we observe steady performance gains as training progresses. Therefore, we report the results at convergence, specifically after 20 epochs of pre-training and 20 epochs of fine-tuning.

### D.2 AUDIO GENERATION

There is a lack of consistency in the metrics used across text-to-audio generation baselines. Some baselines, such as Liu et al. (2023b) and Huang et al. (2023a), employ the VGGish network (Hershey et al., 2017) to compute the Fréchet Distance, while others, like Liu et al. (2023c), utilize the PANNs network (Kong et al., 2019), and still others rely on OpenL3 embeddings, such as Evans et al. (2024b). Additionally, some baselines use the LAION CLAP network (Wu et al., 2023b) to compute the CLAP score, whereas others use the Microsoft CLAP network (Elizalde et al., 2023). To further complicate matters, different baselines often report varying results in various publications. To address these discrepancies, we recalculated all metrics for the baselines using their publicly released checkpoints under identical evaluation configurations. Our method significantly outperforms the baselines across all metrics, except for the Fréchet Distance, where it is slightly behind Make-An-Audio 2 (Huang et al., 2023a). Nevertheless, our user study, detailed in the main paper, indicates that GenAu is generally preferred over Make-An-Audio 2.

| Data Source | # pairs |
|---|---|
| AudioSet | 339.4k |
| VGGSounds | 126.9k |
| Freesounds | 262.3k |
| BBC Sound Effects | 31.2k |
| YouTube Videos | 57.0M |
| ACAV-100M | |
| VideoCC | |
| YTTEMP1B | |
| HDVila-100M | |
| AutoReCap | 761.1k |
| AutoReCap-XL | 57.0M |

Table 6: Overview of the employed dataset sources and audio clips counts for each of them.

Table 7: Comparative overview of the main audio-language datasets.

| Dataset | # Text-Audio Pairs | Duration (h) | Text source |
|---------|-------------------|--------------|-------------|
| AudioCaps | 52,904 | 144 | Human |
| Clotho | 5,929 | 37 | Human |
| MACS | 3,537 | 10 | Human |
| WavText5K | 4,072 | 23 | Online raw-data |
| SoundDescs | 32,979 | 1,060 | Online raw-data |
| LAION-Audio-630K | 633,526 | 4,325 | Online raw-data |
| WavCaps | 403,050 | 7,567 | Processed raw-data |
| AutoReCap | 761,113 | 8,763 | Automatic re-captioning |
| AutoReCap-XL | 57M | 123,500 | Automatic re-captioning |
| AutoReCap-XL-Raw | 327.3M | - | Automatic re-captioning |

Table 8: Audio Evaluation Criteria

| Criterion | Description |
|-----------|-------------|
| Realism | Which audio is more realistic? In other words, is more likely to be a result of a real action. |
| Quality | Which audio has better quality, regardless of the realism of the audio. Please note that some audio may have background noise, which should not be confused with low quality. |
| Prompt Alignment | Considering the prompt to generate the audio is "A sewing machine operating as a machine motor hisses loudly in the background", which audio better follows the given prompt? |
| Overall Preference | Considering the realism, quality, and prompt alignment of the audio, which audio do you prefer more overall? The prompt is: "A sewing machine operating as a machine motor hisses loudly in the background." |

## D.3 USER STUDY

Each user study reported in this paper involved 5 different participants, yielding a total of 1000 responses per study. Samples were selected from the AudioCaps test split, specifically choosing the top 200 samples with the longest text prompts and sampling 50 for each study to enhance the likelihood of obtaining more complex audio scenarios. To minimize discrepancies between baselines, we fix the seed and other sampling parameters across all experiments.

During the user study, participants were initially presented with two audio clips from the compared baselines and asked to judge which one sounded more realistic. They were then prompted to choose the audio they believed had better quality. Next, after showing the prompt used to generate the audio, participants were asked to select the clip that most faithfully followed the prompt. Finally, they were asked to choose their overall preferred audio clip. A screenshot of the user study interface is included in Fig. 12, and the questions posed to the annotators are detailed in Tab. 8.

## E TRAINING AND INFERENCE DETAILS

### E.1 AUTOCAP

AutoCap introduces 6.2 million new parameters on top of the frozen HTSAT audio encoder and the base BART model. These parameters include 4.7M for the Q-Former, 0.9M for embedding layers, and 0.6M for projection layers. The Q-Former employs 256 learnable tokens, a hidden dimension of 256, 8 attention heads, and 2 hidden layers.

Table 9: Qualitative comparison of captioning results on the AudioCaps dataset. See the *Website* for qualitative results accompanied by the respective audio.

| Method | Caption |
|---|---|
| Ground Truth | *A man talking as ocean waves trickle and splash while wind blows into a microphone* |
| Ours | *A man speaks as wind blows and water splashes* |
| CoNeTTE | *A man is speaking and wind is blowing* |
| ENCLAP | *A man is speaking and wind is blowing* |
| Ground Truth | *An adult male speaks, birds chirp in the background, and many insects are buzzing* |
| Ours | *Birds chirp in the distance, followed by a man speaking nearby, after which insects buzz nearby* |
| CoNeTTE | *A man speaking with birds chirping in the background.* |
| ENCLAP | *Birds are chirping and a man speaks* |
| Ground Truth | *A telephone dialing tone followed by a plastic switch flipping on and off* |
| Ours | *A telephone dialing followed by a series of plastic clicking then plastic clanking before plastic thumps on a surface* |
| CoNeTTE | *A telephone ringing followed by a beep.* |
| ENCLAP | *A telephone dialing followed by a series of electronic beeps* |
| Ground Truth | *A running train and then a train whistle* |
| Ours | *A train moves getting closer and a horn is triggered* |
| CoNeTTE | *A train horn blows and a steam whistle is blowing* |
| ENCLAP | *A train running on railroad tracks followed by a train horn blowing as wind blows into a microphone* |
| Ground Truth | *A child is speaking followed by a door moving* |
| Ours | *A child speaks followed by a loud crash and a scream* |
| CoNeTTE | *A woman speaking followed by a door opening and closing.* |
| ENCLAP | *A young girl speaks followed by a loud bang* |
| Ground Truth | *Water splashing as a baby is laughing and birds chirp in the background* |
| Ours | *A baby laughs and splashes, and an adult female speaks* |
| CoNeTTE | *A baby is laughing and people are talking.* |
| ENCLAP | *A baby laughs and splashes in water* |
| Ground Truth | *Leaves rustling in the wind with dogs barking and birds chirping* |
| Ours | *Birds chirp in the distance, and then a dog barks nearby* |
| CoNeTTE | *A dog is barking and a person is walking.* |
| ENCLAP | *Birds chirp and a dog barks* |
| Ground Truth | *Tapping followed by water spraying and more tapping* |
| Ours | *Some light rustling followed by a clank then water pouring* |
| CoNeTTE | *A toilet is flushed and water is running.* |
| ENCLAP | *A faucet is turned on and runs* |

We train the audio captioning model using the Adam optimizer, starting with a learning rate of $10^{-4}$ in stage 1, and reducing to $10^{-5}$ in stage 2. The training was completed over 9 hours on eight A100 80GB GPUs. Although our model is training with 10-second audio clips, we observed qualitatively that it generalizes well to short audios, such as 1-2 second audio clips.

### E.2 GENAU

We employ the LAMB optimizer for our audio generation model, setting the learning rate at 0.005 with a cosine schedule, and incorporating a weight decay of 0.1 and a dropout rate of 0.1. The small model variant is trained for 210k steps with a batch size of 2,048, while the large model variant is trained for 220k steps with a batch size of 3,072. The large model is trained over 48 hours on 48 A100 80GB GPUs, and the small model on 32 GPUs. Ablation studies are conducted on eight A100 80GB GPUs using a batch size of 512. We further condition the model on the training dataset with a conditioning dataset ID. For generation, we utilize the AudioCaps dataset ID as it is the most reliable dataset.

Table 10: Ablation of different FIT architectural variations in terms of patch size number of latent tokens and adopted text encoders on the AudioCaps dataset.

| Tokens | Patch size | FLAN-T5 | CLAP | FD ↓ | FAD ↓ | IS ↑ |
|--------|-----------|---------|------|------|-------|------|
| 256 | 1 | ✓ | ✓ | **16.45** | **1.29** | **10.26** |
| 256 | 1 | | ✓ | 17.41 | 1.39 | 10.0 |
| 256 | 1 | ✓ | | 20.47 | 1.86 | 8.89 |
| 384 | 1 | | ✓ | 17.41 | 1.39 | 10.0 |
| 192 | 1 | | ✓ | 18.0.1 | 2.01 | 8.91 |
| 128 | 1 | | ✓ | 25.56 | 1.77 | 7.49 |
| 256 | 2 | ✓ | ✓ | 18.53 | 1.70 | 9.0 |

Table 11: Ablation of different 1D-VAE designs on audio generation on the AudioCaps dataset.

| Channels | Recon. loss | FAD ↓ | FD ↓ | IS ↑ |
|----------|-------------|-------|------|------|
| 64 | 0.159 | **1.29** | **16.45** | **10.26** |
| 128 | 0.107 | 1.43 | 16.78 | 10.11 |
| 256 | **0.064** | 1.80 | 18.63 | 9.43 |

## F  DISCUSSION WITH CONCURRENT WORK

### F.1  TEXT-CONDITIONED AUDIO GENERATION

Recently, Stable Audio Open (Evans et al., 2024c) introduced a 1.32B-parameter model capable of generating variable-length stereo audio clips at 44.1 kHz. This model leverages a latent diffusion approach with a DiT (Peebles & Xie, 2023a) as its diffusion backbone, similar to prior work such as Make-An-Audio 2 (Huang et al., 2023a). In contrast, GenAu employs a FiT architecture. In Tab. 5, we show the superiority of our FiT-based approach over DiT by showing that GenAu-S is consistently preferred over a 937M-parameter DiT-based baseline (Make-An-Audio 2 Huang et al. (2023a)) when trained on comparable data settings (*i.e.*without recaptioning) at a smaller scale (493M parameters). Additionally, Stable Audio Open proposes directly encoding audio clips using a variational autoencoder (VAE) with a ResNet-like architecture, which is particularly effective for higher-resolution audio generation. In contrast, our work adopts previous approaches (Huang et al., 2023a; Liu et al., 2023c) and uses a Mel-spectrogram representation due to its simplicity. GenAu, being a latent model, can readily benefit from improved latent audio representations, such as those employed by Stable Audio Open.

### F.2  AUDIO CAPTIONING

A concurrent work, SOUND-VECAPS (Yuan et al., 2024), and Auto-ACD (Sun et al., 2024), propose prompting a pretrained large language model with multimodal information. SOUND-VECAPS utilizes visual captions generated by a pretrained visual captioner (Wang et al., 2024b) alongside audio captions from a pretrained audio captioner, ENCLAP (Kim et al., 2024b), to produce more complex captions, showing significant improvements in the downstream task of audio generation. This aligns with our approach of incorporating visual captions in the audio captioning task. However, unlike these methods, which rely solely on pretrained models, we integrate visual information directly into the training process of the audio captioner. This enables a more dynamic and context-aware incorporation of visual information in the audio captioning task.

Additionally, there has been a recent trend toward training large audio-language models (Ghosh et al., 2024b; Kong et al., 2024; Gong et al., 2024b; Deshmukh et al., 2024b) and utilizing them for audio captioning in zero-shot settings. While promising in the pursuit of general-purpose models, their reported results on audio captioning remain inferior to state-of-the-art automatic audio captioning (AAC) methods. Consequently, we opt to train a dedicated AAC model, AutoCap, to achieve the highest-quality captions for our proposed dataset, AutoReCap.

Table 12: Ablation of token replication factors for the HTSAT embeddings extraction procedure of (Mei et al., 2023a) on the AudioCaps test split. Larger token replication factors consistently improve performance due to the related compute increase in the downstream model.

|  | Tokens Count | Replication Factor | CIDEr | BLEU1 | BLEU4 | ROUGE$_L$ |
|---|---|---|---|---|---|---|
| HTSAT-BART | 32 | 1x | 73.7 | 68.6 | 25.0 | 49.7 |
| HTSAT-BART | 256 | 8x | 74.4 | 69.7 | 26.0 | **49.8** |
| HTSAT-BART | 1024 | 32x | **76.6** | **71.5** | **26.3** | **49.8** |
| AutoCap | 32 | 1x | 81.9 | 71.7 | 28.9 | 51.3 |
| AutoCap | 1024 | 32x | **82.7** | **72.5** | **29.3** | **52.0** |

# G ADDITIONAL RESULTS

In this section, we present additional results which are complemented by our *Website*.

## G.1 ADDITIONAL AUDIO CAPTIONING EVALUATION

In Tab. 9 we show qualitative results of the captions produced by our method and compare them with state-of-the-art AAC methods. See the *Website* for qualitative results accompanied by the original audio. While ENCLAP (Kim et al., 2024b) and CoNeTTE (Étienne Labbé et al., 2023) tend to produce short captions, our method produces the most descriptive captions, capturing the most amount of elements from the ground truth audio, an important capability to allow high-quality audio generation (Shi et al., 2020).

## G.2 ADDITIONAL AUDIO GENERATION EVALUATION

In this section, we report additional evaluation results and ablations on the task of audio generation.

In Tab. 10, we evaluate fundamental architectural choices in the design of our scalable FIT model. When removing either the Flan-T5 or CLAP encodings, we notice a steady reduction in all metrics. When increasing the number of latent tokens we also notice a steady improvement in performance as more compute is allocated to the model. Similarly, increasing the patch size to 2 results in a performance decrease under all metrics due to the reduced amount of allocated computation.

In Tab. 11, we ablate the 1D-VAE bottleneck size in terms of reconstruction loss and performance of a subsequently trained latent audio diffusion model, in terms of FAD, FD, and IS. Similarly to the phenomenon observed in the image and video generation domain (Gupta et al., 2023; Esser et al., 2024), we observe that a larger number of channels allocated to the latent space results in lower reconstruction losses, but making the latent space more complex, hindering generation quality. We adopt 64 1D-VAE channels for all our experiments.

## G.3 ADDITIONAL HTSAT EMBEDDING EXTRACTION EVALUATION

We perform a series of ablations on HTSAT-BART Mei et al. (2023a) employing different variants of the procedure of Mei et al. (2023a) for the extraction of HTSAT embeddings (see Appx. B.1). We consider HTSAT output tokens of shape $32 \times 768$ after the averaging operation over the frequency dimension of Mei et al. (2023a), and apply different token repetition factors to produce embeddings with 32 tokens (no token repetition), 256 tokens (8x token repetition) and 1024 tokens (32x token repetition following Mei et al. (2023a)). For completeness, we perform the same ablation on our AutoCap, using as input to the Q-Former 32 tokens (no token repetition) and 1024 tokens (32x token repetition). Training hyperparameters of AutoCap are modified to match HTSAT-BART Mei et al. (2023a) for the purpose of the ablation.

We followed the training procedure of Mei et al. (2023a) and report evaluation results on the AudioCaps test split for the last obtained checkpoint in Tab. 12 and Fig. 13. As the ablation shows, the token replication operation consistently improves model performance. We attribute this finding to the increased computation in the downstream model caused by it and consequently adopt the best

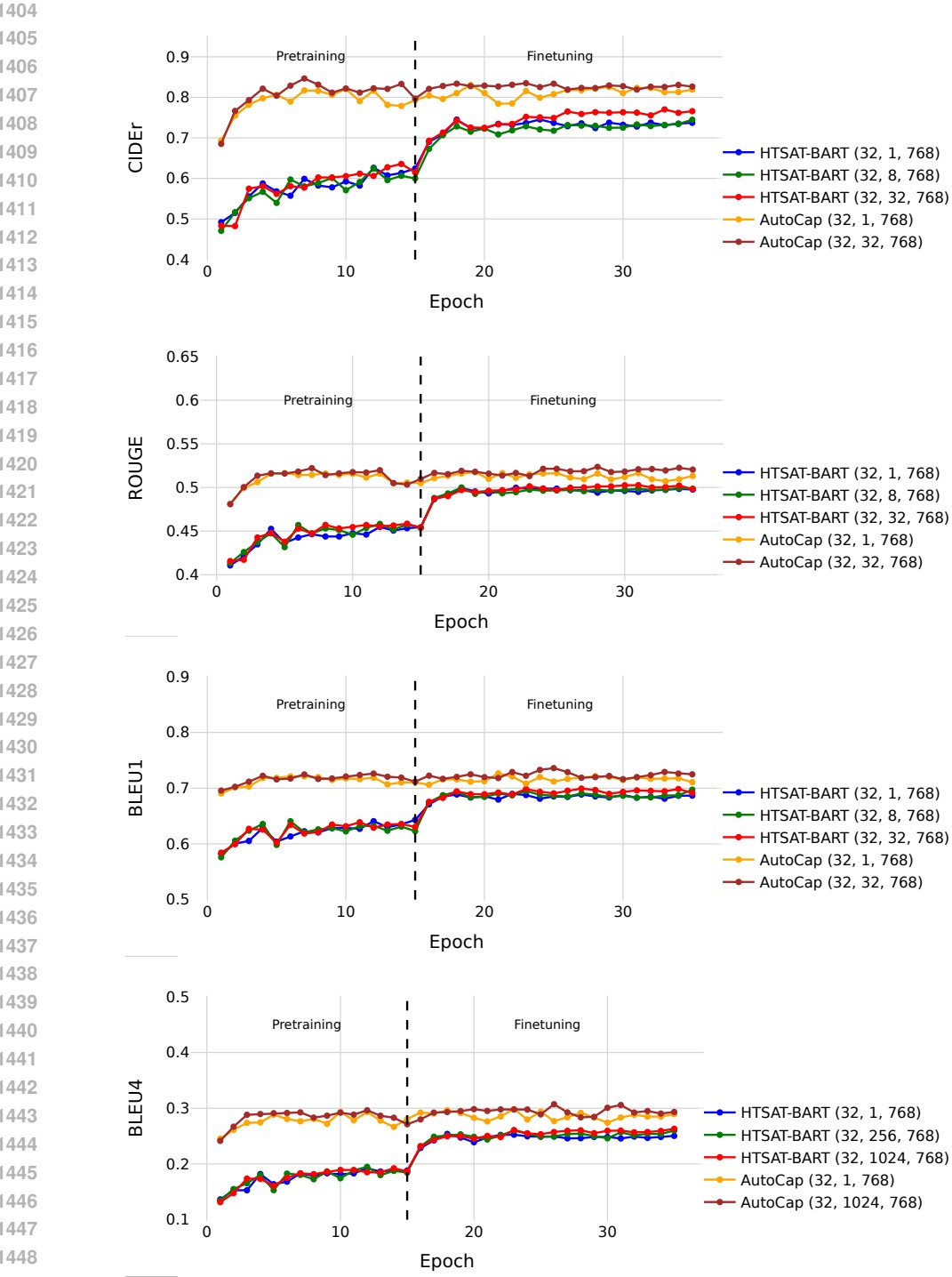

Figure 13: Ablation of token replication factors for the HTSAT embeddings extraction procedure of (Mei et al., 2023a) on the AudioCaps test split for the HTSAT-BART (Mei et al., 2023a) and our AutoCap model.

performing 32x token replication embeddings extraction procedure of Mei et al. (2023a) throughout our work.

