# OpenReview forum: "Taming Data and Transformers for Audio Generation"
_ICLR.cc/2025/Conference — Submitted to ICLR 2025_

### Official Review · Reviewer_bns3 · 2024-10-19

**Soundness:** 4
**Presentation:** 3
**Contribution:** 3
**Rating:** 6
**Confidence:** 5

**Summary:**

This paper introduces a high-quality and efficient audio captioning model, named AutoCap, which demonstrates improvements in generation quality while being four times faster than current state-of-the-art (SoTA) models. The authors further present AutoReCap-XL, a large-scale audio-language dataset comprising 57 million ambient clips paired with automatically generated captions. Additionally, the paper proposes an audio generation model named GenAu, which also achieves SoTA performance.

**Strengths:**

1. The authors employ a Q-Former structure to compress audio representations from the pretrained HTSAT model, which significantly reduces system inference time. Additionally, to develop a more stable system, the paper adopts a two-stage training strategy: in stage one, only the Q-Former and CLAP-projector are trained on weakly labeled samples, while in stage two, the system is fully fine-tuned on the AudioCaps dataset.
2. The paper collects the majority of its data from videos, filtering out ambient clips that lack automatic transcripts. This is a straightforward yet interesting method to exclude speech and music content.
3. According to experimental results, both the captioning and generation systems achieve state-of-the-art (SoTA) performance.

Overall, this paper proposes an interesting approach by leveraging external metadata (e.g., captions or titles from visual information) to enhance audio captioning performance. Moreover, the introduction of the Q-Former module effectively reduces complexity and inference time. Finally, the authors present AutoReCap-XL, a dataset significantly larger than any existing audio-language datasets.

**Weaknesses:**

1. The paper lacks a detailed pipeline explaining how captions are analyzed and how speech or music-related content is filtered out. The authors should provide more clarification on this method.
2. The proposed audio captioning system appears to require external "metadata" to generate captions. However, the definition of this "metadata" is ambiguous throughout the paper. The authors should offer more details on what this metadata consists of and how it is used. If this metadata is not readily accessible from raw audio, it limits the applicability of the system to real-world scenarios.
3. According to Table 1, the comparison indicates that the proposed AutoCap does not achieve the best performance on several metrics, especially when only audio is used as input data.
4. For the audio generation system, the proposed model largely follows the architecture from Huang et al. (2023), incorporating the 1D-VAE and LDM modules. This feels more like an engineering effort, where the existing system is applied and scaled to larger datasets. Moreover, the model uses both FLAN-T5 and CLAP for conditional input, whereas previous models generally employ only one text encoder to achieve satisfactory results. The authors should explain why both encoders are necessary, along with experimental comparisons showing if using only one encoder leads to a drop in performance.
5. The paper lacks an evaluation of the effectiveness of the proposed AutoReCap-XL dataset.

Overall, the paper presents an interesting approach to improving the captioning system using external metadata. However, the explanation of certain methods and strategies needs more detail. Additionally, the metadata required by the captioning system may not be easily accessible from raw audio, making it difficult to apply the system to general tasks. The proposed generation model seems more like an engineering effort rather than a novel contribution. Lastly, the large-scale dataset introduced in the paper requires experimental validation to demonstrate its effectiveness in audio-language tasks. I am willing to change the score if the author can fulfil the experiments for the proposed two systems as well as the dataset.

**Questions:**

1. What does the rejection rate in line 282 refer to? Is the rejection related to downloads?

2. In Section 3.2, the authors mention that they use the captions of each video from the video-captioning model in Panda-70M. Does this imply that the "metadata" is essentially raw captions generated by another captioning system?

3. Can the authors provide more details about the collected video datasets? Specifically, what is the average length of each video sample?

4. How does GenAu perform if only one text encoder (either FLAN-T5 or CLAP) is used?

5. What is the performance of GenAu when trained on the proposed AudioReCap-XL dataset?

6. Could the authors provide some demo examples of GenAu?

**Details Of Ethics Concerns:**

No ethics concerns required.

---

> ### Author Response · Authors · 2024-11-23
>
> We appreciate the reviewer's feedback. Below, we address the reviewer’s main concerns. Due to the character limit, we split our response into two messages.
>
> >W1: The paper lacks a detailed pipeline explaining how captions are analyzed and how speech or music-related content is filtered out. The authors should provide more clarification on this method
>
> In Fig. 2 and Sec. 3.2, we give a high-level overview of the dataset collection pipeline. We include in Appendix Sec. A.1 the initial data selection strategy, Sec. A.2 describes the initial phase for filtering music and speech, and Sec A.3 describes our post-processing strategy for filtering speech and music. In summary, we assume that YouTube audio segments from a given video with no transcriptions are more likely to contain ambient sounds (no speech or music). After selecting such segments and captioning them with AutoCap, we further filter audio segments with captions containing keywords related to music and speech (such as talking, singing, etc).
>
> >W2: The proposed audio captioning system appears to require external "metadata" to generate captions. However, the definition of this "metadata" is ambiguous throughout the paper. The authors should offer more details on what this metadata consists of and how it is used. If this metadata is not readily accessible from raw audio, it limits the applicability of the system to real-world scenarios.
> >W3: According to Table 1, the comparison indicates that the proposed AutoCap does not achieve the best performance on several metrics, especially when only audio is used as input data.
>
> We build a captioning method specifically targeted at producing a large-scale audio generation dataset instead of aiming for raw audio captioning. We specify the metadata used for our final training in L386-388 under training details, which are video captions and the title of the video/audio clip. Such metadata is available for a significantly large set of 57M clips, showing the applicability of the system in real-world scenarios. Lastly, **AutoCap without metadata** (trained with AC + CL + WC) in Tab. 1 **still outperform baselines** in most metrics, except BLUE4 (achieving 28.1 compared with 28.9), and CIDEr (80.0 compared with 80.6), showing that the method is applicable even to specific use cases where only raw-audio is present.
>
>
> >W4: For the audio generation system, the proposed model largely follows the architecture from Huang et al. (2023), incorporating the 1D-VAE and LDM modules. This feels more like an engineering effort, where the existing system is applied and scaled to larger datasets. Moreover, the model uses both FLAN-T5 and CLAP for conditional input, whereas previous models generally employ only one text encoder to achieve satisfactory results. The authors should explain why both encoders are necessary, along with experimental comparisons showing if using only one encoder leads to a drop in performance.
>
> In contrast to Huang et al. (2023) which uses a DiT, our audio generator uses an FiT architecture. We show, for the first time, the superiority of this architecture in the audio domain in Tab.5 where  GenAU-S (493M params) **trained without recaptioning** is preferred over a bigger model Make-an-audio 2 (937M params) trained under similar data settings. We followed previous work [1, 2] in employing both FLAN-T5 and CLAP embeddings, which have been found to bring improvements in the literature as cited in L460-462. We do not claim novelty in the usage of dual text encoders and we refer to prior studies for analyses on the text encoder.
>
> >W5: The paper lacks an evaluation of the effectiveness of the proposed AutoReCap-XL dataset.
>
> We provided AutoReCap-XL as an additional contribution to the paper, complementing our contributions in audio captioning, generation, and dataset collection. However, due to the significant size of the dataset (57M clips) and our limited resources, we are not able to train GenAU on the full dataset. Instead, we provide numerous evaluations that validate the usage of synthetic captions using a dataset size comparable to previous work:
>
> 1- **In Fig. 3 (right)**, we show that the model scales well with the use of more data with synthetic captions. This strongly suggests that AutoReCap-XL can bring compounding improvements.
>
> 2- **In Tab. 3**, we show that GenAU-S training with synthetic captions outperforms significantly GenAU-S without recaptioning in out-of-distribution settings,  improving IS by **60.4%** and confirming the importance of data recaptioning.

---

> ### Author Response · Authors · 2024-11-23
>
> >Q1: What does the rejection rate in line 282 refer to? Is the rejection related to downloads?
>
> Rejection rate refers to the percentage of videos that are rejected when filtering the dataset from speech and music. In other words, it represents the percentage of videos that do not contain ambient sounds and rather speech and music from randomly selected videos.
>
> >Q2: In Section 3.2, the authors mention that they use the captions of each video from the video-captioning model in Panda-70M. Does this imply that the "metadata" is essentially raw captions generated by another captioning system?
>
> The metadata used for training are a caption for the visual modality and the title of the input clip, as specified in L386-388 under training details. We emphasize that the captions used for the metadata are specifically for the visual modality and do not incorporate information from the audio input.
>
> >Q3: Can the authors provide more details about the collected video datasets? Specifically, what is the average length of each video sample?
>
> Appendix Sec. A includes statistics for the collected dataset. Fig. 9 shows the distribution of the clip lengths in AutoRecap-XL, with over 40% of the videos being at least 10 seconds long.
>
> >Q4: How does GenAu perform if only one text encoder (either FLAN-T5 or CLAP) is used?
>
> We have not conducted such an ablation, as it is well-established in the audio generation literature [1, 2] that a dual text encoder strategy offers improvements. The same observation has been validated in the image generation [3] and video generation [4] domains Since the use of dual text encoders is not a primary contribution of our work, we have not run an ablation on this design.
>
> >Q5: What is the performance of GenAu when trained on the proposed AudioReCap-XL dataset?
>
> Please refer to our response to Weakness 5.
>
> >Q6: Could the authors provide some demo examples of GenAu?
>
> We would like to respectfully draw the reviewer’s attention to the attached website in the supplementary material, which contains a large quantity of qualitative results of GenAU, as well as dataset samples, and qualitative comparisons on both the generation and captioning tasks.
>
> **Final message:** We sincerely thank the reviewer for their comments and useful feedback. We believe we have addressed all the questions and concerns raised. If the reviewer has any additional questions or finds any of our responses unsatisfactory, we would be more than happy to revisit and address them further.
>
> **References**
>
> 1- Make-An-Audio 2: Temporal-Enhanced Text-to-Audio Generation, 2023
>
> 2- AudioLDM 2: Learning Holistic Audio Generation with Self-supervised Pretraining, CVSSP 2023
>
> 3- Scaling Rectified Flow Transformers for High-Resolution Image Synthesis, 2024
>
> 4- Movie Gen: A Cast of Media Foundation Models, 2024

---

> ### Author Response · Authors · 2024-11-24
>
> Dear Reviewer bns3,
> We would like to renew our availability to further address any remaining concerns that may be present in these last days of discussion.
> We are looking forward to receiving your feedback.

---

> ### Comment · Reviewer_bns3 · 2024-11-24
>
> Thank you for the author's response. While the author has addressed my concerns to some extent, the absence of an evaluation for the proposed AudioReCap-XL dataset remains a significant issue. Despite its large scale and the challenges associated with training or evaluation, the dataset should be tested to ensure its reliability and address any potential issues prior to publication. Otherwise, the dataset cannot be claimed as a contribution and should not be used by the public. Additionally, the author states that the proposed system utilizes a FiT structure, which differs from the Make-an-Audio2 system. However, it is unclear whether any novel improvements have been made to the original FiT structure introduced in the prior CV paper. As it stands, the approach seems to involve incorporating various "new" techniques into the existing system to achieve better performance, rather than presenting a fundamentally innovative framework. I look forward to your further clarification on these points.

---

> ### Author Response · Authors · 2024-11-25
>
> We sincerely thank the reviewer for engaging in the discussion. Below we address the reviewer's remaining concerns.
>
> **GenAU contribution**
>
> We demonstrated the feasibility of adapting the  FiT architecture to **latent** audio diffusion models and demonstrated the **scalability and superiority** of the FiT architecture in the audio generation task. This is **significantly important** since in contrast to other domains (e.g. image and video) which benefit significantly from scaling, previous and concurrent UNet-based and DiT-based methods [1, 2] have reported **poor scalability**. For instance, AudioLDM-2 [1] reported that their **712M** model (AudioLDM 2-Full-Large), achieve **worse results** than their **346M** model in **all metrics**, except REL (3.80 compared with 3.77). Similarly, a concurrent **DiT-based** method, EzAudio [2] reported a **worse CLAP** score, and only marginal improvement of  **0.26%** in IS when comparing their largest to the smaller model. This is in contrast to our proposed FiT architecture which shows its superior scalability (**see Fig. 2, left**).
>
>
> While we do not claim novelty on the FiT architecture itself, we would like to argue that introducing an existing architecture to a new task, providing insights into its optimal configuration, and validating its effectiveness and scalability is an important contribution which has has been deemed novel and influential in many prior work [3, 4].
>
> **AudioReCap-XL**
>
> We have contributed AutoReCap, a significant dataset of 761.1k text-audio pairs, and have **verified its effectiveness** in scaling audio generation (Fig. 3, right, Tab. 3, and Tab. 5). Our analysis highlights that even this dataset alone brings substantial value to the audio generation community.
>
> In an effort to provide the community with an even more significant dataset, we scaled the dataset collection approach to provide AutoReCap-XL, which follows the **same collection strategy** of AutoReCap, and we have provided several analyses on the dataset distribution in Sec. A.
>
> We agree with the reviewer and have acknowledged in the limitation section **L1111-1112** that further validation of AutoReCap-XL would be desirable, but we also believe it is valuable to release such data to the community as it currently suffers from a lack of large-scale dataset, a point raised by other reviewers (**4Kh5, DQFx**), and it is reasonable to suggest that conclusions on a subset of the dataset (AutoReCap) would likely transfer to the XL version (AutoReCap-XL) as the same collection strategy is followed. While it is not feasible to train GenAU with AutoReCap-XL within the discussion period deadline, if the paper gets accepted, we will include this experiment in the camera-ready version for completeness.
>
> We would like to respectfully reiterate our core contributions in the paper:
>
>   1- AutoCap: a novel audio captioning architecture that achieves state-of-the-art performance.
>
>   2- GenAU: an adaptation of the FiT architecture to the audio generation task. We demonstrate its effectiveness and scalability in this new setting while achieving state-of-the-art performance.
>
>   3- A scalable data collection pipeline for ambient audio that we used to collect AutoReCap, of which we extensively evaluated the benefits in the audio generation task.
>
> Finally, we offer AutoReCap-XL as an additional contribution, providing the community with access to a large-scale dataset of 57M audio-text pairs. We believe this is valuable and decided to provide it as a supplement alongside the extensively evaluated AutoReCap dataset, despite its more limited evaluation.
>
> We sincerely thank the reviewer for their engagement in the discussion and hope that our responses have adequately addressed their concerns. Should the reviewer have any additional questions or concerns, we would be glad to revisit them.
>
> **Reference**
>
> 1- AudioLDM 2: Learning Holistic Audio Generation with Self-supervised Pretraining, TASLP 2024.
>
> 2- EzAudio: Enhancing Text-to-Audio Generation with Efficient Diffusion Transformer, 2024.
>
> 3- Scalable Diffusion Models with Transformers, ICCV 2023.
>
> 4- An Image is Worth 16x16 Words: Transformers for Image Recognition at Scale, ICLR 2021

---

> ### Comment · Reviewer_bns3 · 2024-11-25
>
> Thanks for the reply from the author. I am happy to increase the score to 6.

---

### Official Review · Reviewer_TUnZ · 2024-10-24

**Soundness:** 2
**Presentation:** 3
**Contribution:** 2
**Rating:** 3
**Confidence:** 5

**Summary:**

The paper proposes a new method to improve audio (sound) generation. The authors first propose AutoCap, a novel method to generate audio captions using auto-regressive models. Next, the authors propose a novel audio generation model which is trained on their generated dataset. The proposed method shows improvement on benchmark datasets.

**Strengths:**

- The paper is well written and well presented. The figures are good and the writing and everything is crisp. It is a nice to read paper.
- The method shows good improvements. Open-sourcing the artifacts in future would help the audio community.
- Th intuitions are good. The fact that good captions can improve audio generation is a good finding and well conveyed. Although I feel some parts are over-claimed which I mention next.
- I don't see many technical flaws with the paper.

**Weaknesses:**

I have several issues with the paper. I will first point out the technical weaknesses:
- Fig. 1 says CLAP Encoder has one token. CLAP uses HTSAT as the audio encoder  which also has intermediate representations. This only means that the authors used the CLS token (or some pooled representations) which is not specified. The authors should also clearly mention "CLAP audio encoder".
- The caption says "We then compact this representation into 4x fewer tokens using a Q-Former (Li et al., 2023a) module.". The figure shows only HTSAT encoder representation is passed to QFormer. The authors should rewrite the caption.
- The claim "4x fewer tokens" just because QFormer was employed in one part is not justified. The authors are also using BART embeddings, etc. Do you mean "4x fewer tokens" compared to your own baseline which does not use a QFormer?
- QFormer has been used for audio earlier [3]. Additionally, it does not seem like you are pre-training the QFormer? Does this mean you are training it E2E? An E2E model as big as QFormer trained on such small datasets is not very sound.
- The authors say fewer tokens but do not talk about the increase in parameter count. I would like to see ore discussion on this please.
- Some audio captioning prior art missing from comparison [1,2].
- I am concerned why only audio clips with "No subtitles" were uses. Is the synchronization of time and subtitle correct? Also, human speech is an important sound and heavily present in audiocaps. If the authors believe not, then I think the paper should refocus on "environmental sounds" and not "audio".
- (Not a big weakness) I feel its important that synthetic data generation pipelines have some verification strategy (human or automatic).
- The freezing and unfreezing paradigm has been heavily employed in prior art in the DCASE challenge. The paper misses some citations. Please see DCASE captioning challenges that use BART like architectures.
- The fact that visual modality helps in audio captioning is not new and has been explored. Comparison or discussion with papers like SOUND-VECAPS [4] or [5] is missing.
- For GenAu, a discussion with Stable Audio is missing. I understand where at some places the authors mention DiT and how GenAu mitigates some limitations of DiT based architectures, but the introduction of time embeddings and the use of attention modules makes it very similar to StableAudio overall. A discussion would do justice.
- Table 4 does not compare with Stable Audio. Same training data + Stable Audio comparison is a must for a fair comparison.
- The table has missing comparison of baselines + their re-captioned data. I struggle to understand where the gains are coming from. The only ablation of GenAU w. U-Net is not sufficient. I also understand computational constraints but a couple of baselines with the data would do more justice to the results.

[1] https://ieeexplore.ieee.org/abstract/document/10448030.
[2] https://arxiv.org/abs/2309.07372.
[3] https://arxiv.org/abs/2406.11768.
[4] https://arxiv.org/abs/2407.04416.
[5] https://arxiv.org/abs/2309.11500.

**Questions:**

My questions are broadly mentioned in the Weaknesses section. I am mostly looking for the "Why?" answers to the choices made that are related to my listed Weaknesses.

---

> ### Author Response · Authors · 2024-11-23
>
> We appreciate the reviewer's feedback. Please find below our detailed responses to the reviewer’s comments. Due to the character limit, we split our response into two messages.
>
> > W1: Fig. 1 says CLAP Encoder has one token. CLAP uses HTSAT as the audio encoder which also has intermediate representations. This only means that the authors used the CLS token (or some pooled representations) which is not specified. The authors should also clearly mention "CLAP audio encoder".
>
> We use the CLAP [1] audio encoder to extract audio embedding (a single token) following the instructions in their Github page [1]. We have clarified this in the updated paper.
>
> > W2: The caption says "We then compact this representation into 4x fewer tokens using a Q-Former (Li et al., 2023a) module.". The figure shows only HTSAT encoder representation is passed to QFormer. The authors should rewrite the caption.
>
> Only the HSAT embedding (1024 tokens) is passed to the Q-Former. Since the CLAP audio embeddings (a single token) and BART metadata embeddings (64 tokens) are limited, we pass them directly to the BART encoder as the figure shows. We have refined the caption to indicate that Q-Former only receives the HTSAT representation.
>
> >W3: The claim "4x fewer tokens" just because QFormer was employed in one part is not justified. The authors are also using BART embeddings, etc. Do you mean "4x fewer tokens" compared to your own baseline which does not use a QFormer?
>
> We compare the input of Q-Former (1024 tokens) with the output (256 tokens) when reporting the 4x reduction. In other words, if Q-Former is not employed, 1024 tokens would be passed to the BART encoder, similar to previous work [4, 5]. We additionally pass metadata embeddings (64 tokens) and CLAP audio embeddings (1 token) directly to the BART encoder. We have refined the caption to make it clear that 4x fewer tokens refer to the HTSAT representation.
>
> >W4: QFormer has been used for audio earlier [3]. Additionally, it does not seem like you are pre-training the QFormer? Does this mean you are training it E2E? An E2E model as big as QFormer trained on such small datasets is not very sound.
>
> We do not pretrain Q-Former, and AutoCap is trained E2E. Our Q-Former is lightweight, containing only 4.7M parameters. We have updated Appendix Sec D.1 with details on the hyperparameters that we use for the Q-Former module. We would like to note that GAMA [2] appeared on arxiv only three months before the ICLR deadline and we consider it as concurrent work.
>
> >W5:
> The authors say fewer tokens but do not talk about the increase in parameter count. I would like to see ore discussion on this please.
>
> AutoCap introduces only a total of 6.2M parameters, with 4.7M for Q-Former, 0.9M for embedding layers, and 0.6 for projection layers. We have updated Appendix Sec D.1 with details on the parameter count.
>
> >W7: I am concerned why only audio clips with "No subtitles" were uses. Is the synchronization of time and subtitle correct? Also, human speech is an important sound and heavily present in audiocaps. If the authors believe not, then I think the paper should refocus on "environmental sounds" and not "audio".
>
> As mentioned in the abstract and frequently throughout the text, our work focuses on **“ambient sounds”** or, in other words, environmental sounds, as the reviewer suggests. Data for this task is more difficult to obtain since most online videos are dominated by speech and music. Additionally, although YouTube automatic transcriptions give accurately synchronized subtitles, the subtitle content is sometimes inaccurate. To address this, we employ an additional keyword-based filtering step to filter speech and music clips from the dataset as mentioned in Fig. 2 and Sec A.3.
>
> >W8: (Not a big weakness) I feel its important that synthetic data generation pipelines have some verification strategy (human or automatic).
>
> We have obtained these captions using our state-of-the-art captioner (AutoCap). We state in the limitations section our agreement with the reviewer that more validation would be desirable and we acknowledge that the captions are not perfect, leaving room for improvement. We plan on releasing the dataset, providing the opportunity to improve the captions in future work.

---

> > ### Author Response · Authors · 2024-11-23
> >
> > >W11: For GenAu, a discussion with Stable Audio is missing. I understand where at some places the authors mention DiT and how GenAu mitigates some limitations of DiT based architectures, but the introduction of time embeddings and the use of attention modules makes it very similar to StableAudio overall. A discussion would do justice.
> >
> > Stable Audio Open uses a DiT as the diffusion backbone, whereas GenAU uses a FiT. The two architectures are fundamentally different since FiT relies on the learnable latent token and ‘read’ and ‘write’ operation which distributes the computation more dynamically between the audio tokens compared to a DiT. We have included in Sec E.1 a discussion comparing our method with Stable-Audio Open. We would like to note that Stable-Audio open is an arxiv preprint that appeared two months before the ICLR deadline and we regard it as a concurrent work.
> >
> > >W12: Table 4 does not compare with Stable Audio. Same training data + Stable Audio comparison is a must for a fair comparison.
> >
> > Stable-Audio Open [3] is an arxiv preprint that appeared two months before the ICLR deadline. It is not computationally feasible to train large-scale baselines with our dataset for ablation purposes. However, we perform an **equivalent comparison**. We have included a comparison of GenAU-S (493M params) trained **without recaptioning** with the previous SOTA method Make-an-audio 2 (937M params) in Tab. 5 (i.e at a similar data setting), showing the superiority of our architecture compared with a DiT. As requested by the reviewer, we also include a comparison of GenAU-L with Stable-Audio Open.
> >
> > | Metric             | FD    | FAD   | IS    | CLAP  |
> > |--------------------|-------|-------|-------|-------|
> > | Stable-Audio Open   | 21.23 | 2.32  | 10.48 | 0.584 |
> > | Ours               | 16.51 | 1.21  | 11.75 | 0.668 |
> >
> > >W13: The table has missing comparison of baselines + their re-captioned data. I struggle to understand where the gains are coming from. The only ablation of GenAU w. U-Net is not sufficient. I also understand computational constraints but a couple of baselines with the data would do more justice to the results.
> >
> > Unfortunately, It is not feasible to train large-scale baselines with our recaptioned data. However, **we perform an equivalent evaluation, showing gains that can exclusively be attributed to our architecture**:
> > - **Architecture:** as shown in Tab. 5, GenAu-S **without Recap dataset** outperformed Make-an-audio-2 when trained at similar data setting (**without audio recaptioning**), confirming the superiority of our architecture.
> > - **Data quality:** In Tab. 3, we show that GenAU-S greatly outperforms **GenAu-S without Recap**, improving IS by **60.4%** and confirming the importance of data recaptioning.
> >
> > >W6: Some audio captioning prior art missing from comparison [1,2].
> >
> > > W9: The freezing and unfreezing paradigm has been heavily employed in prior art in the DCASE challenge. The paper misses some citations. Please see DCASE captioning challenges that use BART like architectures.
> >
> > > W10: The fact that visual modality helps in audio captioning is not new and has been explored. Comparison or discussion with papers like SOUND-VECAPS [4] or [5] is missing.
> >
> > We thank the reviewer for highlighting the missing citations. Our paper includes over 120 citations referencing prior work, including studies exploring the use of visual modality in audio captioning (L122–123). However, we acknowledge that some relevant citations were missing. We have updated the paper to include the citations suggested by the reviewer as well as a discussion on concurrent work in Sec E.2.
> >
> > **Final message:** We sincerely appreciate the reviewer’s time and effort in evaluating our work. While we acknowledge the feedback regarding missing citations (4, 6, 9, 10), questions about Q-Former usage (1, 2, 3, 5), and the request for a comparison with Stable-Audio Open (11, 12, 13)—an arXiv preprint published two and a half months before the ICLR deadline—we believe these points pertain to refinements and additional analyses rather than fundamental weaknesses in our methods. We kindly ask the reviewer to reconsider our contributions across three key areas: dataset collection, captioning, and generation. We would sincerely appreciate the reviewer to highlight any weakness calling for rejection if any.
> >
> > **References:**
> >
> > 1- Large-scale Contrastive Language-Audio Pretraining with Feature Fusion and Keyword-to-Caption Augmentation, ICASSP 2023 (https://github.com/LAION-AI/CLAP)
> >
> > 2- GAMA: A Large Audio-Language Model with Advanced Audio Understanding and Complex Reasoning Abilities, ACL 2024
> >
> > 3- Stable Audio Open, arxiv 2024
> >
> > 4- WavCaps: A ChatGPT-Assisted Weakly-Labelled Audio Captioning Dataset for Audio-Language Multimodal Research, TASLP 2023
> >
> > 5- CoNeTTE: An efficient Audio Captioning system leveraging multiple datasets with Task Embedding, TASLP 2024

---

> > > ### Comment · Reviewer_TUnZ · 2024-11-24
> > > **Further Clarification**
> > >
> > > Thank You for your response! I have further queries.
> > >
> > > > We compare the input of Q-Former (1024 tokens) with the output (256 tokens) when reporting the 4x reduction.
> > >
> > > Q-Former has 1024 tokens as input? I thought HTSAT last layer token length is 64? What is 1024 then? have the authors confused embedding dimension with token length? and when you say Q-Former has 256 tokens as output do you mean that is what your length of query tokens is? As far as I remember the general setup has 32 tokens and 256 hidden dimension?
> > >
> > > On another note, the number query tokens is adjustable and just a hyper-parameter? What made the authors come to 256 (incase that is the correct number).
> > >
> > > Can the authors also please highlight why they use both htsat and CLAP? CLAP uses htsat and the authors could have just used intermediate representations from CLAP for best of both worlds?
> > >
> > > ---------
> > >
> > > On other note, I am convinced with other responses to other questions except probably the prior art part. I acknowledge Stable Audio Open is a two month old arxiv paper (before ICLR submission), however https://arxiv.org/abs/2402.04825 by the Stable Audio authors has been up since February and published in ICML and the authors of the paper have released the corresponding and related codebase on github much earlier.
> > >
> > > Finally, if the paper is accepted, I request the authors to add some baselines + their re-captioned data experiment for the final version to strengthen the argument and the dataset contribution and ideally show that their architecture is superior.

---

> > > > ### Author Response · Authors · 2024-11-25
> > > >
> > > > We sincerely thank the reviewer for engaging in the discussion. Below we address the reviewer’s comments.
> > > >
> > > > >Q-Former has 1024 tokens as input? I thought HTSAT last layer token length is 64? What is 1024 then? have the authors confused embedding dimension with token length? and when you say Q-Former has 256 tokens as output do you mean that is what your length of query tokens is? As far as I remember the general setup has 32 tokens and 256 hidden dimension?
> > > >
> > > > We respectfully note that HTSAT encodes a **10-second audio clip** sampled at 32kHz into 1024 tokens, each with a dimension of 527, as mentioned in the training details of HTSAT [1], “**The shape of the output feature map is (1024, 527) (C=527).**”  We use the representation from the last hidden layer, with dimensions 1024 × 768, which we passed to the Q-Former model. Using 256 learnable queries with a hidden dimension of 256, the Q-Former generates a feature map of size 256 × 256.
> > > >
> > > >
> > > > >On another note, the number query tokens is adjustable and just a hyper-parameter? What made the authors come to 256 (incase that is the correct number).
> > > >
> > > > Yes, the number of query tokens is adjustable. We conducted an ablation to decide on the number of learnable queries. Using **128 learnable queries** led to underperformance compared to our base model, achieving a **CIDEr of 77.1** (compared with 83.2). Additionally, using **1024 queries** is almost 4x slower for training and inference while reporting small improvements (**CIDEr of 84.1**). Overall, we found that using 256 tokens strikes a good balance. We will include these ablations in the final version.
> > > >
> > > >
> > > >
> > > > >Can the authors also please highlight why they use both htsat and CLAP? CLAP uses htsat and the authors could have just used intermediate representations from CLAP for best of both worlds?
> > > >
> > > > We appreciate the reviewer’s suggestion. CLAP trains its internal HTSAT [1] representation with a contrastive learning objective on a relatively noisy dataset (LAION-Audio-630K). As a result, its learned representation appears to be inferior to supervised training, underperforming HTSAT in audio classification [1].
> > > >
> > > > Instead, inspired by [2], we combine a pretrained HTSAT representation (1024 tokens) with a pretrained CLAP audio embedding (1 token). As we report in the **ablation Tab. 4**, incorporating CLAP improves the CIDEr score, while maintaining similar performance in METEOR and SPICE.
> > > >
> > > >
> > > >
> > > > >On other note, I am convinced with other responses to other questions except probably the prior art part. I acknowledge Stable Audio Open is a two month old arxiv paper (before ICLR submission), however https://arxiv.org/abs/2402.04825 by the Stable Audio authors has been up since February and published in ICML and the authors of the paper have released the corresponding and related codebase on github much earlier.
> > > >
> > > > We are glad that we have addressed the reviewer’s other concerns. We would like to respectfully note that a quantitative comparison with Stable Audio 1.0 [3] is infeasible for two reasons:
> > > >
> > > > - Stable Audio 1.0 was trained on an internal dataset and its pretrained checkpoints are **not publicly available**. Moreover, the authors of Stable Audio 1.0 [3] employed a **different evaluation setup** than all other existing audio generators [5,6], using the OpenL3 network to compute the Frechét Distance and a ‘feature fusion’ variant of CLAP to compute the CLAP score. This makes it infeasible to compute **standard metrics** on Stable Audio. We touched on this topic in **L1133-1134**.
> > > > - We consider the possibility of retraining Stable Audio 1.0 [3] using their codebase. The authors of [3] reported using **64 A100 GPUs** to perform 640,000 training steps. However, 64 A100 40GB nodes on AWS (8xp4d.24xlarge) cost **6291$ per day** [7]. It would be unreasonably expensive to run such training **for multiple days**.
> > > >
> > > > As a result, we instead used their **commercial API** [4], which was previously hosting Stable Audio 1.0, to provide a qualitative comparison in the **website** in the supplementaries.
> > > >
> > > >
> > > > > Finally, if the paper is accepted, I request the authors to add some baselines + their re-captioned data experiment for the final version to strengthen the argument and the dataset contribution and ideally show that their architecture is superior.
> > > >
> > > > We appreciate the reviewer’s request. Stable Audio Open is trained for 14 days on 64 A100 GPUs.  64 A100 40GB nodes on AWS cost **6291$** per day [7], giving a total estimated cost of **88,598$** to retrain Stable Audio Open. Make an Audio 2 [5] uses 8 A100 GPUs for a cost of **786$** per day. Training this model for the 1.2M steps performed by the authors requires an estimated 5.53 days with a total cost of **4349$**.  While it is not feasible to run this experiment by the discussion period deadline, should the paper be accepted, we will include a comparison with Make-an-Audio 2 in the proposed settings in the camera-ready version.
> > > >
> > > > We would be happy to address other concerns if any remain.

---

> > > > > ### Author Response · Authors · 2024-11-25
> > > > >
> > > > > **References**
> > > > >
> > > > > [1] HTS-AT: A Hierarchical Token-Semantic Audio Transformer for Sound Classification and Detection, ICASSP 2022
> > > > >
> > > > > [2] EnCLAP: Combining Neural Audio Codec and Audio-Text Joint Embedding for Automated Audio Captioning, 2024
> > > > >
> > > > > [3] Fast Timing-Conditioned Latent Audio Diffusion, ICML 2024
> > > > >
> > > > > [4] Stable Audio: https://stableaudio.com/
> > > > >
> > > > > [5] Make-An-Audio 2: Temporal-Enhanced Text-to-Audio Generation, 2023
> > > > >
> > > > > [6] AudioLDM 2: Learning Holistic Audio Generation with Self-supervised Pretraining, CVSSP 2023
> > > > >
> > > > > [7] https://aws.amazon.com/ec2/instance-types/p4

---

> ### Comment · Reviewer_TUnZ · 2024-11-25
> **Response to response**
>
> Thank You for your response.
>
> > We respectfully note that HTSAT encodes a 10-second audio clip sampled at 32kHz into 1024 tokens, each with a dimension of 527, as mentioned in the training details of HTSAT [1], “The shape of the output feature map is (1024, 527) (C=527).”
>
> I would like to respectfully clarify a misunderstanding on the authors end. For (1024, 527) -- 527 is the number of audioset classes and 1024 is the dimension of the embedding. Thus, the authors have probably used the latent output of the classification layer for their experimental setup which is not sound. No paper does it, neither for audio or vision. The representations in this latent will have very less useful information and is steered towards multi-label classification as is a collapsed embedding passed through classification layers.
>
>  What should have been used is the latent representation before the classification layer, which is 64 x 1024.

---

> > ### Comment · Reviewer_4Kh5 · 2024-11-26
> > **Issue regarding HTS-AT**
> >
> > I went through the discussion over HTS-AT, and I feel that this is a critical issue I overlooked that the authors should address this. Even I am looking forward to the response from the authors.
> >
> > Reiterating what reviewer TUnZ mentioned:
> > Latent representation for HTS-AT before the classification layers are 64 x 1024 for HTS-AT Base and 64 x 2048 for HTS-AT large.

---

> ### Author Response · Authors · 2024-11-27
>
> Dear Reviewer TUnZ and 4Kh5, we sincerely thank you for providing your feedback and engaging in the discussion. We are happy to clarify the point regarding HTS-AT embeddings extraction and usage.
>
> **What is the dimensionality of the HTS-AT latent representation?**
>
> We apologize for the confusing reference in our previous response. With reference to Sec. 2.1.2 in [1], HTS-AT encodes a 10s audio signal into a latent representation before the classification layers (referred to as latent tokens in Fig. 1 in [1]) of shape (T/(8*P), F/(8*P), 8D) = **(32, 2, 768)**, where T=1024, F=64, D=96, P=4 (See Sec 3.1.1 in [1]).
>
> **Which HTS-AT latent space do we use?**
>
> We confirm that we use the customary latent representation **before** the classification layer. **We do not use the classification layer output.**
>
> We closely follow [2] codebase in extracting HTS-AT embeddings. Although not explicitly mentioned in [2] manuscript, [2] first encodes the audio signal using HTS-AT, producing a latent representation before the classification layer of shape (32, 2, 768), referred to as “latent tokens” in Fig. 1 of [2]. It then performs averaging over the second dimension (corresponding to the frequency dimension), obtaining a representation of shape (32, 1, 768), i.e. 32 tokens with 768 channels. As a final step, [2] repeats each token 32 times, obtaining a representation of shape (32, 32, 768) that is flattened into (1024, 768), i.e. 1024 tokens with 768 channels. This representation is referred to as “fine_grained_embeddings” in the codebase of [2] and several other codebases such as CLAP [8]. Several other works employ this “fine_grained_embeddings” representation [3, 4, 5, 6, 7]. We perform further studies of the HTS-AT representation of [2] to show the beneficial effects of using the fine-grained representation.
>
> **Is the token replication operation of [2] necessary?**
>
> We perform a series of ablations on HTSAT-BART [2] employing different variants of the procedure of [2] for the extraction of HTS-AT embeddings. We start with HTS-AT output tokens after the averaging operation of shape (32, 1, 768), and apply different token repetition factors to produce embeddings with 32 tokens (no token repetition), 256 tokens (8x token repetition), and 1024 tokens (32x token repetition following [2]). For completeness, we perform the same ablation on our model, using as input to the Q-Former 32 tokens (no token repetition) and 1024 tokens (32x token repetition).
> We followed [2] training procedure and we report evaluation results on AudioCaps test split on the last checkpoint in the following table. Please note that training hyperparameters of AutoCap are modified to match HTSAT-BART [2] for the purpose of the ablation.
>
> | Model         | Embeddings Shape | Token Repetition Factor | CIDEr | BLEU1 | BLEU4 | ROUGE |
> |---------------|------------------|-------------------------|-------|-------|-------|-------|
> | HTSAT-BART    | (32, 1, 768)     | 1x                      | 73.7  | 68.6  | 25.0  | 49.7  |
> | HTSAT-BART    | (32, 8, 768)     | 8x                      | 74.4  | 69.7  | 26.0  | 49.8  |
> | HTSAT-BART    | (32, 32, 768)    | 32x                     | 76.6  | 71.5  | 26.3  | 49.8  |
> | AutoCap       | (32, 1, 768)     | 1x                      | 81.9  | 71.7  | 28.9  | 51.3  |
> | AutoCap       | (32, 32, 768)    | 32x                     | 82.7  | 72.5  | 29.3  | 52.0  |
>
> Please refer to **Fig. 13** in the updated manuscript for a better understanding of the effect of token replication, where we reported the performance across various epochs in the pretraining and finetuning phases. **As the ablation shows, the token replication operation of [2] consistently improves model performance**. We believe this finding can be attributed to the increased computation in the downstream model caused by it. Although the improvements with fine-grained representation are less pronounced in our model, we observed that the fine-grained representation consistently outperforms the 32 tokens representation (see Fig. 13). Consequently, we adopt the best performing 32x token replication embedding extraction procedure of [2] throughout our work. We also remark that with the chosen Q-Former design and the HTS-AT representation of [2], **our audio captioning model provides state-of-the-art performance (see Table 1).**
>
> We sincerely thank Reviewer TUnZ and 4Kh5 for giving us the chance to discuss the embedding extraction procedure of [2] and to enrich the manuscript with deeper insights and ablations related to it. We updated the manuscript with the discussion and ablation results provided in this answer (see Appx. B.1, Appx. G.3). We hope this answer clarifies all of the Reviewer’s remaining concerns and we remain available if any further concern is present.

---

> > ### Author Response · Authors · 2024-11-27
> >
> > **References**
> >
> > [1] HTS-AT: A Hierarchical Token-Semantic Audio Transformer for Sound Classification and Detection, ICASSP 2022
> >
> > [2] WavCaps: A ChatGPT-Assisted Weakly-Labelled Audio Captioning Dataset for Audio-Language Multimodal Research, TASLP 2023
> >
> > [3] Advancing Multi-grained Alignment for Contrastive Language-Audio Pre-training, ACM MM 2024
> >
> > [4] Parameter Efficient Audio Captioning with Faithful Guidance Using Audio-Text Shared Latent Representation, ICASSP 2024
> >
> > [5] EmotionCaps: Enhancing Audio Captioning Through Emotion-Augmented Data Generation
> >
> > [6] Investigating Passive Filter Pruning for Efficient CNN-Transformer Audio Captioning, MLSP 2024
> >
> > [7] Multi-grained Correspondence Learning of Audio-language Models for Few-shot Audio Recognition, ACM MM 2024
> >
> > [8] Large-scale Contrastive Language-Audio Pretraining with Feature Fusion and Keyword-to-Caption Augmentation, ICASSP 2023

---

> > ### Comment · Reviewer_TUnZ · 2024-11-29
> > **Response**
> >
> > Thank You for your response. I am not very convinced with the response. In your first response you said  `(1024, 527)` which made sense from an architecture perspective for CLAP-base, however it looked like it was an experimental flaw. Now I am assuming that was said my mistake in the rebuttal.
> >
> > Now that you say  `(32, 2, 768)`, which is true for CLAP-tiny and not base. But I have another problem which is -- usually what happens at this layer according to the original implementation and CLAP on HF is this:
> >
> > https://github.com/huggingface/transformers/blob/main/src/transformers/models/clap/modeling_clap.py#L997
> >
> > This would lead to `(64, 768)` for the tiny model (which I said in my response) -- but now the authors mention that `(32, 2, 768)` --> `(32, 32, 768)` -- which is not the conventional and widely adopted setup. The authors next show some results based on the token replication (32, 32) -- but I am not sure why would anyone scientifically do it (purely ablative and only probably done in [2] -- apologize for not reading [2] in detail but I don't see the exact lines which point towards this experimental setup).
> >
> > This  `(32, 32, 768)` now matches 1024 when flattened in a non-conventional was -- as said by the authors. In conclusion:
> > - The response now has a potential tiny and base conflict from the original response which tried to say that 1024,527 was the output of the model
> > - The methods used, even if sound, drift from the usual setup. And it is hard for me to find the setup.
> >
> > I will maintain my score and request the authors to have a closer look at their experimental setup for future revisions.

---

> > > ### Author Response · Authors · 2024-11-30
> > >
> > > We thank the reviewer for their response. Please allow us to address below the reviewer’s raised points.
> > >
> > > > In your first response you said (1024, 527) which made sense from an architecture perspective for CLAP-base, however it looked like it was an experimental flaw. Now I am assuming that was said my mistake in the rebuttal.
> > >
> > > We sincerely apologize again for the confusion caused by the misleading reference in our second-to-last response regarding the **(1024, 527)** output of HTSAT. While we mentioned that “we use the representation from the last hidden layer”, we acknowledge that the explanation was unclear and potentially misleading.
> > >
> > > However, **in our previous response and the updated manuscript, we provided a detailed and accurate explanation** of how the "fine_grained_representation" we used in our model was obtained. The reviewer found this representation to be **sound**, despite their doubts about its effectiveness.
> > >
> > > >The response now has a potential tiny and base conflict from the original response which tried to say that 1024,527 was the output of the model.
> > >
> > > We clarify that we adopted [1] implementation, which uses HTSAT configuration as mentioned in the HTSAT paper [2]. We have not mentioned the HTSAT configuration from CLAP, neither tiny nor base.
> > >
> > > > The methods used, even if sound, drift from the usual setup.
> > >
> > > We clarify that this setup has been used in previous peer-reviewed work HTSAT-BART [1], on which we based our codebase, and that such representation has been adapted in various other peer-reviewed papers [4, 5, 6, 7].
> > >
> > > > [...] and it is hard for me to find the setup.
> > >
> > > > [...] I don't see the exact lines which point towards this experimental setup)
> > >
> > > We invite the reviewer to refer to the source code of HTSAT-BART [1]  (https://github.com/XinhaoMei/WavCaps/blob/a5a9649ce305d7fe82cfcf5d6a4a12f03df9ef1e/captioning/models/htsat.py#L840C9-L842C75), (https://github.com/XinhaoMei/WavCaps/blob/a5a9649ce305d7fe82cfcf5d6a4a12f03df9ef1e/captioning/models/htsat.py#L953-L955) and to the detailed embeddings extraction procedure reported in our previous answer and updated manuscript. Unfortunately, the authors of [1] do not discuss this setup in their manuscript.
> > >
> > > > I am not sure why would anyone scientifically do it (purely ablative and only probably done in [2]
> > >
> > > While **it is outside of the scope of this work to prove the effectiveness of a technique employed in a widely-adopted [4, 5, 6, 7] peer-reviewed publication [1]**, we note that:
> > >
> > > - There is no reason not to adopt a setup that can be experimentally proven to work better (as shown in our previous response) than other baselines setup.
> > >
> > > - Even without token replication, we would achieve the state-of-the-art performance, as reported in our previous response.
> > >
> > >
> > > **Closing Remarks**
> > >
> > > We would like to reiterate the large scope of the contributions contained in the paper which span a wider context than the focus of the current discussion:
> > >
> > > **1- AutoReCap:** A scalable data collection pipeline for ambient audio that we used to collect AutoCap, a large-scale dataset for which we have verified its effectiveness in scaling the performance in the downstream task of audio generation (See Fig.3 right, Tab. 3). As a supplement, we have scaled our collection pipeline to contribute AutoReCap-XL, a 57M audio-text pairs, almost 90x larger than previous datasets.
> > >
> > > **2- GenAU:** A scalable audio generation model that achieves state-of-the-art performance for which we verified its scalability (see Fig. 3 left, Tab. 3, Tab. 5). This is significant since previous UNet-based generators (e.g AudioLMD2 [8]) and DiT-based generators (e.g EzAudio [9]), have reported that their largest model either underperforms or achieves very marginal improvements over their smaller model).
> > >
> > > **3- AutoCap:** A novel audio captioning model that achieves state-of-the-art performance (See Tab. 1).
> > >
> > > We hope that our answer addresses the reviewer’s concern. From our discussion with the reviewer, we understand that their remaining concern pertains to the representation that we used for our captioning model. We would like to note that our approach is sound (as acknowledged by the reviewer) and has been adopted in previous work [1, 3, 4, 5, 6, 7], has been ablated (Fig. 13), and yields state-of-the-art performance in audio captioning. We believe **this concern can not warrant rejection in the broad context of our contributions in audio captioning, as well as dataset collection and audio generation**.
> > >
> > > We again sincerely thank the reviewer for their time and for engaging in the discussion and we confirm our availability to discuss any other concern.

---

> > > > ### Author Response · Authors · 2024-11-30
> > > >
> > > > **References**
> > > >
> > > > [1] WavCaps: A ChatGPT-Assisted Weakly-Labelled Audio Captioning Dataset for Audio-Language Multimodal Research, TASLP 2023
> > > >
> > > > [2]  HTS-AT: A Hierarchical Token-Semantic Audio Transformer for Sound Classification and Detection, ICASSP 2022
> > > >
> > > > [3] Advancing Multi-grained Alignment for Contrastive Language-Audio Pre-training, ACM MM 2024
> > > >
> > > > [4] Parameter Efficient Audio Captioning with Faithful Guidance Using Audio-Text Shared Latent Representation, ICASSP 2024
> > > >
> > > > [5] EmotionCaps: Enhancing Audio Captioning Through Emotion-Augmented Data Generation, 2024
> > > >
> > > > [6] Investigating Passive Filter Pruning for Efficient CNN-Transformer Audio Captioning, MLSP 2024
> > > >
> > > > [7] Multi-grained Correspondence Learning of Audio-language Models for Few-shot Audio Recognition, ACM MM 2024
> > > >
> > > > [8]AudioLDM 2: Learning Holistic Audio Generation with Self-supervised Pretraining, TASLP 2024.
> > > >
> > > > [9] EzAudio: Enhancing Text-to-Audio Generation with Efficient Diffusion Transformer, 2024.

---

> > > > > ### Author Response · Authors · 2024-12-02
> > > > >
> > > > > Dear reviewer TUnZ,
> > > > >
> > > > > As today is the last day for reviewers to submit comments to the authors, we would greatly appreciate your feedback regarding the remaining concern and are happy to address any additional concerns that you may have.

---

### Official Review · Reviewer_DQFx · 2024-10-31

**Soundness:** 3
**Presentation:** 3
**Contribution:** 2
**Rating:** 6
**Confidence:** 4

**Summary:**

Introduce two new models. First, we propose AutoCap, a high-quality and efficient automatic audio captioning model. By using a compact audio representation and leveraging audio metadata, AutoCap substantially enhances caption quality Second, the work proposes GenAu, a scalable transformer-based audio generation architecture that is scaled up to 1.25B parameters. Using AutoCap to generate caption clips from existing audio datasets.

**Strengths:**

* The paper presents a simple pipeline to label audio data in order to generate larger data than ever before.
* The paper shows the new dataset improves the quality of trained models.
* Train a SOTA model using the new dataset.
* The authors say they will release the dataset which could be a good contribution to the community.

**Weaknesses:**

* In scenarios where metadata lacks detail, audio captioning may struggle to disambiguate sounds accurately. The model also tends to falter in
capturing the temporal relationships between sounds and differentiating foreground from background
noises.
* Fine-tuned on AudioCaps, which contains a limited vocabulary of 4,892 unique words. The limited vocabulary of the paired texts, even though extensive, hampers the model’s ability to accurately generate audio for long and detailed prompts.

* The proposed dataset, AutoReCap-XL, is substantial in size but features a constrained vocabulary of
only 4,461 unique words. Furthermore, despite its potential as a significant contribution, this dataset has not
yet been extensively analyzed for caption accuracy or performance in downstream tasks.

**Questions:**

* Didn't train baselines on the new dataset to show the proposed architecture is actually superior. I would have liked to see the baselines trained on the 57M example dataset. This would clearly show if the better performance of the proposed method is due to architecture or just the scaling of the dataset.
* Is there a way to verify the quality of the dataset in terms of captioning? assuming the community adopts the use of this new data how are you to ensure the data is of high quality? maybe some human evaluation would be in place.
* Please move the limitations section into the main body of the paper.

---

> ### Author Response · Authors · 2024-11-23
>
> We thank the reviewer for their feedback.  Below, we address the reviewer’s comments.
>
> > W1: In scenarios where metadata lacks detail, audio captioning may struggle to disambiguate sounds accurately. The model also tends to falter in capturing the temporal relationships between sounds and differentiating foreground from background noises.
>
> In Tab. 1, we show state-of-the-art audio captioning performance. Without metadata (trained with AC + CL + WC) our method still outperforms baselines in most metrics, except BLUE4 (achieving 28.1 compared with 28.9), and CIDEr (80.0 compared with 80.6). Additionally, in the attached Website, we show that our method has better temporal understanding than the previous approaches. We agree with the reviewer that no perfect audio captioner exists. For this reason, we highlight this observation in our Limitations section.
>
> >W2: Fine-tuned on AudioCaps, which contains a limited vocabulary of 4,892 unique words. The limited vocabulary of the paired texts, even though extensive, hampers the model’s ability to accurately generate audio for long and detailed prompts.
>
> Audio captioning performance is limited by the available datasets with ground truth captions, with AudioCaps being the largest. This is a key challenge faced by most previous work. Using the visual modality as an additional input helps improve the diversity of the generated words, producing words unseen in the training dataset. However, as we discussed in the limitation section B.2, we acknowledge that there is still room for improvement.
>
> >W3: The proposed dataset, AutoReCap-XL, is substantial in size but features a constrained vocabulary of only 4,461 unique words. Furthermore, despite its potential as a significant contribution, this dataset has not yet been extensively analyzed for caption accuracy or performance in downstream tasks.
>
> We introduce AutoReCap-XL as an additional contribution after contributing a state-of-the-art audio captioner, a state-of-the-art audio generation model, and AutoReCap, for which we evaluated its benefits in audio generation. As an additional contribution, we introduce AutoReCap-XL, which is significant in size, featuring 57M audio-text pairs. We expect the data scaling trends shown in Fig. 3 to transfer to this dataset. While we have conducted limited analysis of this dataset in Sec. A, as noted in the conclusion (Sec. 5), we leave more extensive analyses for future work.
>
> >Q1: Didn't train baselines on the new dataset to show the proposed architecture is actually superior. I would have liked to see the baselines trained on the 57M example dataset. This would clearly show if the better performance of the proposed method is due to architecture or just the scaling of the dataset.
>
> We conducted evaluations to show that our architecture is better regardless of the data. In Tab. 5, where we show that GenAu-S without Recap outperformed the strongest available baseline, Make-an-audio-2, when trained at a similar data setting (without audio recaptioning), confirming the superiority of our architecture. This evaluation setting is common in text-to-audio evaluation. Running several large-scale baselines on a large-scale dataset would be an unnecessarily expensive way of evaluating the same architectural advantage.
>
> >Q2: Is there a way to verify the quality of the dataset in terms of captioning? assuming the community adopts the use of this new data how are you to ensure the data is of high quality? maybe some human evaluation would be in place.
>
> We verify the quality of AutoReCap by means of the downstream task of text-to-audio generation. Our out-of-distribution evaluation (Tab. 3) confirms that recaptioning with AutoCap significantly improves the quality of the generated audio, achieving **60.4%** better IS. We have verified the effect of scaling audio generation models with this synthetic dataset in Fig 3 (right).
>
> >Q3: Please move the limitations section into the main body of the paper.
>
> We have included a summarized discussion of the method’s limitations in the updated main manuscript according to the reviewer’s suggestion, and we will keep a detailed limitation section in the Appendix.
>
> **Final Message:** We sincerely thank the reviewer for their feedback. We believe we have addressed all concerns but are happy to revisit any additional questions if needed.

---

> > ### Author Response · Authors · 2024-11-25
> >
> > Dear Reviewer DQFx, we would like to renew our availability to further address any remaining concerns that may be present in these last days of discussion. We are looking forward to receiving your feedback.

---

> > > ### Author Response · Authors · 2024-12-02
> > >
> > > Dear reviewer DQFx,
> > >
> > > We hope that our answers resolved you concerns. As today is the last day for reviewers to submit comments to the authors, we would greatly appreciate your feedback regarding your remaining concerns and are happy to address any additional concerns that you may have.

---

### Official Review · Reviewer_4Kh5 · 2024-11-03

**Soundness:** 2
**Presentation:** 3
**Contribution:** 2
**Rating:** 6
**Confidence:** 5

**Summary:**

This paper proposes an audio captioning method called AutoCap, an audio generation model called GenAu and an audio dataset called AutoReCap-XL.

**Strengths:**

- The illustrations in the paper are clear and it is well written.
- No evident flaws.
- AutoReCap-XL will be great asset for the audio research community if open-sourced.

**Weaknesses:**

- The authors shared the audio samples from Stable-Audio 1.0[7] but the comparison is missing in results table.
- Typo in table 5, last row "Quality" column: "-".
- The performance improvement when compared to increase in number of parameters in GenAu is marginal.
- Recently, Large Audio Language models [1,2,3,4] are being employed for audio captioning task, but the authors don't compare AutoCap to these baselines which in my opinion should be an important comparison.
- Inconsistent use of word "Q-Former" and "Qformer".
- Why do authors not show generation results of GenAu without Recap dataset or baseline + Recap in normal settings? It should be an important ablation to show where the performance gain is coming from.
- The CLAP text conditioner is confusing, the text encoder employed by CLAP [5,6] is BERT. What do authors mean when they say that they have used CLAP text encoder, do they use the text encoder weights from CLAP's checkpoint?
- There should be clear distinction between CLAP and EnCLAP. Authors mention CLAP but cite EnCLAP.
- The authors propose to use video caption and title as metadata, how sensitive is the output to the accuracy and relevance of metadata? Have the authors performed any ablation to show the robustness of this method?
- It would be interesting to analyse the distribution of sounds across various classes of non-ambient sounds in AutoReCap-XL. Have the authors explored any distribution patterns, and is there a noticeable bias toward any particular class as there is a lot of filtering while preparing the dataset?
- In section A.2 authors mention "Given that AutoReCap was trained for 10-second audio", I think there is a typo and it should be "AutoCap" instead of "AutoReCap" as the later one is the dataset.
- I want to understand what is the novelty of GenAu when compared to other models? In my opinion all other baselines and other audio generative models are **almost** capable of generating more or less similar audios with lesser parameters.

References

[1] GAMA: A Large Audio-Language Model with Advanced Audio Understanding and Complex Reasoning Abilities

[2] Listen, Think, and Understand

[3] Audio Flamingo: A Novel Audio Language Model with Few-Shot Learning and Dialogue Abilities

[4] Pengi: An Audio Language Model for Audio Tasks

[5] ENCLAP: COMBINING NEURAL AUDIO CODEC AND AUDIO-TEXT JOINT EMBEDDING FOR AUTOMATED AUDIO CAPTIONING

[6] CLAP : LEARNING AUDIO CONCEPTS FROM NATURAL LANGUAGE SUPERVISION

[7] Stable Audio Open

**Questions:**

Please see weakness section.

---

> ### Author Response · Authors · 2024-11-23
>
> We appreciate the reviewer's feedback. Please find below our detailed responses to the reviewer’s comments. Due to the character limit, we split our response into two messages.
>
> > W1: The authors shared the audio samples from Stable-Audio 1.0[7] but the comparison is missing in results table.
>
> Since Stable-Audio Open [1] generates at a different setting than all other baselines (stereo, 44khz), we decided not to include it in the table of compassion. We would also like to note that Stable-Audio open is an arxiv preprint that was published two and half months before the ICLR deadline. As requested by the reviewer, we included a comparison of our work with Stable-Audio Open in Tab. 4, showing that GenAU-L outperforms Stable-Audio open in all metrics.
> | Metric             | FD    | FAD   | IS    | CLAP  |
> |--------------------|-------|-------|-------|-------|
> | Stable-Audio Open   | 21.23 | 2.32  | 10.48 | 0.584 |
> | Ours               | 16.51 | 1.21  | 11.75 | 0.668 |
>
> > W2: The performance improvement when compared to increase in number of parameters in GenAu is marginal.
>
> We would like to clarify the subject of the question with the reviewer.
> If the question concerns a comparison between our method and baselines with respect to the number of parameters, we refer the reviewer to our subsequent answer to W12.
>
> If the question instead concerns scalability properties of the proposed architecture with respect to model size, We respectfully disagree with the statement. We discuss model scaling in L509-L513 and we have verified the effectiveness of model scaling in three different evaluation settings:
>
> 1- **Scaling trend (Fig. 3):** Fig. 3 (left) in the paper shows a clear trend of improvements in both FD and IS when increasing the parameter count. Notably, the largest model (1.25B) outperformed the smallest (123M) by **56.3%** in IS and **36.6%** in FD under identical settings.
>
> 2- **Out-of-Distribution Evaluation (Tab. 3):**  In out-of-distribution evaluation in Table. 3, the 1.25B model has **20.4%** improvement in IS and **5.5%** in CLAP score over the 493M model.
>
> 3- **User Study (Tab. 5):** In our user study in Table. 5, the 1.25B model achieved **61.20%** preference in realism, and prompt alignment compared to the 493M model.
>
> We disagree that such improvements are **marginal.**
>
> > W4: Recently, Large Audio Language models [1,2,3,4] are being employed for audio captioning task, but the authors don't compare AutoCap to these baselines which in my opinion should be an important comparison.
>
> We followed recent AAC methods [2, 3] in our evaluation protocol to compare with baselines trained in similar data settings. Large Audio Language models perform captioning in zero-shot settings. While promising, they still underperform current SOTA AAC methods. Moreover, comparing these models using automatic evaluation metrics is not entirely fair, as such metrics are heavily influenced by the captioning style used in the ground truth. We evaluated GAMA—the state-of-the-art Audio LLM—using their most capable model (GAMA-IT) on the AudioCaps test dataset with their default prompt, 'Describe the audio.' Our results show that GAMA-IT reports significantly lower scores across all metrics compared to our model.
>
> | Metric               | SPIDER | SPICE | CIDEr | BLUE1 | BLUE4 | Rouge |
> |----------------------|--------|-------|-------|-------|-------|-------|
> | GAMA                | 35.1   |  5.4  | 16.5  | 20.4  | 12.2  | 19.1  |
> | Ours (audio only)    | 49.7   | 19.0  | 80.4  | 73.1  | 28.1  | 52.0  |
>
>
> Please note that GAMA only reported a SPICE score of 18.5 using 500 random samples from AudioCaps, a different evaluation setting than all previous AAC methods which uses AudioCaps test dataset. We have included in Sec. E.2 a discussion on Large Audio Language models.
>
> > W6: Why do authors not show generation results of GenAu without Recap dataset or baseline + Recap in normal settings? It should be an important ablation to show where the performance gain is coming from.
>
> **We do!** We discussed data recaptioning in L501-L506.  We verified our methods in two axes:
> - **Architecture:** as shown in Tab. 5, **GenAu-S without Recap dataset** outperformed Make-an-audio-2 when trained at similar data setting (**without audio recaptioning**), confirming the superiority of our architecture.
> - **Data quality:** In Tab. 3, we show that GenAU-S greatly outperforms **GenAu-S without Recap**, improving IS by **60.4%** and confirming the importance of data recaptioning. We have also included in the website in the supplementary a qualitative comparison between **GenAu-S without Recap** and GenAU-S, where the improvements in the quality and prompt following are obvious. For example, in the “Motorcycle starting and taking off” sample, the baseline without recaptioning does not follow the prompt well.

---

> > ### Author Response · Authors · 2024-11-23
> >
> > > W7: The CLAP text conditioner is confusing, the text encoder employed by CLAP [5,6] is BERT. What do authors mean when they say that they have used CLAP text encoder, do they use the text encoder weights from CLAP's checkpoint?
> >
> > We closely followed most previous work in employing the CLAP [4] text encoder to condition audio generation [5, 6]. In short, yes, we used CLAP’s pretrained checkpoint to extract CLAP embedding following [5, 6]. For the CLAP evaluation metrics, we use AudioLDM evaluation kit [7]. We have clarified this aspect in the updated manuscript.
> >
> > > W9: The authors propose to use video caption and title as metadata, how sensitive is the output to the accuracy and relevance of metadata? Have the authors performed any ablation to show the robustness of this method?
> >
> > **We have!** We reported in Tab. 1 the results of AutoCap when using only audio (second to last row). Even without metadata (i.e. audio only), we still outperform baselines in most metrics, except BLUE4 where we (achieving 28.1 compared with 28.9), and CIDEr (80.0 compared with 80.6).
> >
> > > W10: It would be interesting to analyse the distribution of sounds across various classes of non-ambient sounds in AutoReCap-XL. Have the authors explored any distribution patterns, and is there a noticeable bias toward any particular class as there is a lot of filtering while preparing the dataset?
> >
> > In Fig. 11, we show a word cloud of the generated captions in AutoReCAP-XL. There is a bias toward common classes such as vehicle engines, and hard surface sounds. However, without filtering, the dataset is dominated by music and speaking sounds (see Fig. 10), an undesirable property for a dataset aimed at sound effects generation.
> >
> > > W12: I want to understand what is the novelty of GenAu when compared to other models? In my opinion all other baselines and other audio generative models are almost capable of generating more or less similar audios with lesser parameters.
> >
> > The reviewer’s statement is in contrast with user study results. Table 5 shows that GenAU-S (**493M params**) when trained **without recaptioning** is preferred over the best-performing baseline Make-an-audio 2 (**937M params**) trained with similar data settings and with 1.90X parameter count, showing the superiority of our audio generator architecture. Furthermore, previous methods (such as AudioLDM2 [8]) have reported poor scalability with their large model (712M) underperforming their smaller model (346M) in most metrics. This is in contrast to our proposed model which benefits from model scalability (see Fig. 3 left)
> >
> > > W2: Typo in table 5, last row "Quality" column: "-".
> >
> > > W5: Inconsistent use of word "Q-Former" and "Qformer".
> >
> > > W8: There should be clear distinction between CLAP and EnCLAP. Authors mention CLAP but cite EnCLAP.
> >
> > > W11: In section A.2 authors mention "Given that AutoReCap was trained for 10-second audio", I think there is a typo and it should be "AutoCap" instead of "AutoReCap" as the later one is the dataset.
> >
> > We appreciate the reviewer pointing out the typos. We have fixed them in the updated version.
> >
> > **Final message:** We sincerely thank the reviewer for their thoughtful feedback which we integrated in our revised version. We believe the points we clarified pertain to refinements rather than core weaknesses and kindly ask the reviewer to reconsider our contributions across three key areas: dataset collection, captioning, and generation. We would sincerely appreciate the reviewer to highlight the weakness calling for rejection if any.
> >
> > **Reference**
> >
> > 1- Stable Audio Open, arxiv 2024
> >
> > 2- EnCLAP: Combining Neural Audio Codec and Audio-Text Joint Embedding for Automated Audio Captioning, ICASSP 2024
> >
> > 3- CoNeTTE: An efficient Audio Captioning system leveraging multiple datasets with Task Embedding, TASLP 2024
> >
> > 4- Large-scale Contrastive Language-Audio Pretraining with Feature Fusion and Keyword-to-Caption Augmentation, ICASSP 2023
> >
> > 5- Make-An-Audio 2: Temporal-Enhanced Text-to-Audio Generation, 2023
> >
> > 6- AudioLDM: Text-to-Audio Generation with Latent Diffusion Model, ICML 2023
> >
> > 7- AudioLDM Audio Generation Evaluation  (https://github.com/haoheliu/audioldm_eval)
> >
> > 8-  AudioLDM 2: Learning Holistic Audio Generation with Self-supervised Pretraining, CVSSP 2023

---

> > > ### Comment · Reviewer_4Kh5 · 2024-11-24
> > > **Response to Official Comment by Authors**
> > >
> > > I thank the authors for their rebuttal, it helped me clarify my doubts. I am increasing by score to 6.

---

> > > > ### Author Response · Authors · 2024-11-25
> > > >
> > > > We greatly appreciate the reviewer’s time and effort in reviewing our work and are delighted to have been able to clarify their concerns.

---

### Meta-Review · Area_Chair_bFdp · 2024-12-22

**Metareview:**

> This paper introduces a high-quality and efficient audio captioning model, named AutoCap, which demonstrates improvements in generation quality while being four times faster than current state-of-the-art (SoTA) models. The authors further present AutoReCap-XL, a large-scale audio-language dataset comprising 57 million ambient clips paired with automatically generated captions. Additionally, the paper proposes an audio generation model named GenAu, which also achieves SoTA performance.

I agree with the authors that the rating of 3 from TUnZ may be a bit harsh post-rebuttal, but even if I were to fully ignore it, there is no champion review. So, weaknesses remain, in particular in disantangling contributions of data and modeling and supporting modeling claims.

This an interesting paper, but it fails a bit short of the scholarship required for ICLR.

**Additional Comments On Reviewer Discussion:**

Authors did a healthy rebuttal and reviewers engaged with it. It was not sufficient to change the ratings of the paper into a clear accept.

---

### Decision · Program_Chairs · 2025-01-22

Reject